# DNAJB1-PRKACA fusion protein-regulated LINC00473 promotes tumor growth and alters mitochondrial fitness in fibrolamellar carcinoma

**Rosanna K. Ma**[1], **Pei-Yin Tsai**[2], **Alaa R. Farghli**[1], **Alexandria Shumway**[1], **Matt Kanke**[1], **John D. Gordan**[3], **Taranjit S. Gujral**[4], **Khashayar Vakili**[5], **Manabu Nukaya**[6], **Leila Noetzli**[7], **Sean Ronnekleiv-Kelly**[6], **Wendy Broom**[7], **Joeva Barrow**[2], **Praveen Sethupathy**[1]*

1 Department of Biomedical Sciences, College of Veterinary Medicine, Cornell University, Ithaca, New York, United States of America, 2 Division of Nutritional Sciences, Cornell University, Ithaca, New York, United States of America, 3 Division of Hematology/Oncology, Helen Diller Family Comprehensive Cancer Center, UCSF, San Francisco, California, United States of America, 4 Human Biology Division, Fred Hutchinson Cancer Center, Seattle, Washington, United States of America, 5 Department of Surgery, Boston Children's Hospital, Boston, Massachusetts, United States of America, 6 Department of Surgery, Division of Surgical Oncology, University of Wisconsin School of Medicine and Public Health, Madison, Wisconsin, United States of America, 7 Alnylam Pharmaceuticals, Cambridge, Massachusetts, United States of America

* pr46@cornell.edu

**Data Availability Statement:** Our RNA-seq datasets on patient tissues, as well as LeGO-473ox and LeGO-Ctl cells, were deposited into the Gene

## Abstract

Fibrolamellar carcinoma (FLC) is a rare liver cancer that disproportionately affects adolescents and young adults. Currently, no standard of care is available and there remains a dire need for new therapeutics. Most patients harbor the fusion oncogene *DNAJB1-PRKACA* (DP fusion), but clinical inhibitors are not yet developed and it is critical to identify downstream mediators of FLC pathogenesis. Here, we identify long noncoding RNA LINC00473 among the most highly upregulated genes in FLC tumors and determine that it is strongly suppressed by RNAi-mediated inhibition of the DP fusion in FLC tumor epithelial cells. We show by loss- and gain-of-function studies that LINC00473 suppresses apoptosis, increases the expression of FLC marker genes, and promotes FLC growth in cell-based and *in vivo* disease models. Mechanistically, LINC00473 plays an important role in promoting glycolysis and altering mitochondrial activity. Specifically, LINC00473 knockdown leads to increased spare respiratory capacity, which indicates mitochondrial fitness. Overall, we propose that LINC00473 could be a viable target for this devastating disease.

## Author summary

Fibrolamellar carcinoma (FLC) is a lethal liver cancer that is characterized by the DNAJB1-PRKACA (DP fusion) oncogene. This disease is particularly challenging to treat as it lacks non-invasive diagnostic biomarkers, known risk factors, and effective

Expression Omnibus (GEO: GSE233148). The RNA-seq data on LeGO-473ox and LeGO-Ctl tumors are accessible through GEO: GSE253465.

**Funding:** This work was supported by funding from the College of Veterinary Medicine Graduate Fellowship (awarded to RKM), Center for Vertebrate Genomics Scholar Award (awarded to RKM), Sandra Atlas Bass Endowment for Cancer Research (awarded to PS), and a Fibrolamellar Cancer Foundation Research Grant (awarded to PS). The authors confirm that the funders had no role in the study design, data collection and analysis, decision to publish, or preparation of the manuscript.

**Competing interests:** Drs. Wendy Broom and Leila Noetzli are employees and shareholders of Alnylam Pharmaceuticals, with multiple patent applications pending. All other authors declare no competing interests.

therapeutic options. Patients with FLC have a median age of 22 years and the prognosis is poor. In this study, we examined FLC tumors from independent patient cohorts and multiple disease models and found consistently elevated levels of a primate-specific long noncoding RNA named LINC00473. By leveraging single-cell analysis on patient tumors, we found that LINC00473 is enriched in tumor epithelial cells. Using multiple FLC models, we show that the DP fusion drives LINC00473 expression and that LINC00473 promotes FLC growth by mitigating cell death. We also show that LINC00473 modulates FLC energetics by increasing glycolysis and altering mitochondrial fitness. Together, this study unveils an important mechanism downstream of the signature DP fusion oncogene in FLC pathogenesis.

## Introduction

Fibrolamellar carcinoma (FLC) is a rare and aggressive type of liver cancer that disproportionately affects young adults, with a median age of onset at 22 years [1–6]. FLC constitutes ~13% of liver cancers for patients less than 40 years old [3]. Unlike adult-onset liver cancers, FLC presents with vague symptoms, lacks diagnostic serum biomarkers, and is not associated with known liver cancer risk factors [4–7]. Surgical resection remains the only curative treatment approach; however, the majority of diagnoses occur at advanced metastatic stages of disease, leading to surgical ineligibility, high rates of recurrence, and poor survival [8–10]. The dire need for therapeutics is underscored by how FLC tumors are resistant to drugs currently used to treat other liver cancers [2,5,8–10].

At the genomic level, FLC tumors are characterized by a somatic, heterozygous ~400kb deletion on chromosome 19, resulting in a fusion gene called *DNAJB1-PRKACA* (DP fusion) [11,12]. The fusion consists of the chaperone-binding domain of heat shock protein 40 (DNAJB1) and the alpha isoform of the catalytic subunit of protein kinase A (PRKACA). The DP fusion is detected in the vast majority of patient tumors and is widely appreciated as the main FLC driver gene [4,13]. Functionally, the DP fusion is sufficient for initiating tumor formation in mice [14,15] and is likely important for tumor maintenance [16], but specific pharmacological targeting of the oncoprotein has proved challenging [17–19]. Therefore, it is critical to identify and target candidate downstream effectors that drive FLC development and progression. Genome-scale studies of FLC tumors have identified many dysregulated genes that are hypothesized to be sensitive to DP fusion activity, and these merit deeper investigation as alternative therapeutic targets [20–28].

Long noncoding RNAs (lncRNAs) are critical regulators of gene expression at the transcriptional and post-transcriptional levels [29–32]. Defined as transcripts over 200 nucleotides in length with little to no protein-coding potential, lncRNAs are frequently expressed in a tissue- and disease-specific manner [33,34]. Functionally, lncRNAs exhibit highly specific cell-type and subcellular expression patterns and bind to proteins, as well as chromatin and RNA, to mediate gene regulation [29,32–37]. Emerging evidence shows that some lncRNAs are abnormally expressed in tumors and play important roles in cell cycle progression, uncontrolled proliferation, anti-apoptosis, dysregulated metabolism and metastasis [38–46]. This suggests that lncRNAs are both targets and regulators of cancer-specific transcriptional programs.

LINC00473 is a putative oncogenic lncRNA in a few cancer types and is part of a reported gene signature of FLC [21,47–51]. Our group previously reported on LINC00473 in FLC, but the findings have been very limited. Specifically, we showed that introducing *DNAJB1-PRKACA* into human embryonic kidney HEK293 cells leads to potent LINC00473 upregulation

[52]. Furthermore, we have also observed that FLC tumor-specific enhancers adjacent to LINC00473 are highly enriched in cAMP response binding protein (CREB) binding motifs [53]. Though intriguing, because CREB is a critical mediator of PKA signaling and has been suggested to regulate LINC00473 in normal physiological and disease settings [47,48,54–57], these studies were highly limited by small sample sizes and subpar cell models available at that time. Therefore, to date, very little is understood about LINC00473 in the context of FLC. Notably, the functional significance of LINC00473 in FLC remains completely unknown.

To bridge the large knowledge gap, we evaluated the function of LINC00473 in FLC progression using genome-wide analyses combined with functional molecular studies in both cell-based and *in vivo* mouse models. In this work, we not only dramatically improve upon our previous report with much larger sample sizes and several independent cohorts but, importantly, also provide a completely new link between LINC00473 and tumor growth in vivo as well as novel findings on LINC00473-mediated regulation of cellular energetics in a much more relevant FLC cell model. We also newly leverage single-cell data to localize LINC00473 to a specific cell type in the tumor.

Our findings implicate a role for LINC00473 in altering mitochondrial activity that may influence pro-tumorigenic cancer cell energetics in FLC. This body of work establishes LINC00473 as one of the most well-characterized FLC marker genes to date. We hypothesize that LINC00473 is a promising diagnostic marker of DP fusion activity and a candidate therapeutic target for FLC.

## Results

### *LINC00473* represents a distinct transcription unit in FLC tumors

In recent years, the LINC00473 locus has been subject to variable annotation across different databases. For example, according to some annotation libraries, the LINC00473 transcript is collapsed into an isoform of a nearby protein-coding gene, *PDE10A*. However, a recent comparative genomic analysis demonstrated that the expression of LINC00473 is driven independently from PDE10A in human neurons derived from induced pluripotent stem cells. The same study confirmed that LINC00473 is present only in higher primates [55]. Several reports have demonstrated that LINC00473 has limited coding potential and that any role of a putative protein from this locus is likely negligible [47,55]. Additionally, others have observed that *PDE10A* was not altered in expression upon LINC00473 modulation in human endometrial stromal cells and adipocytes [54,58]. Taken together, these findings strongly indicate that LINC00473 is an independent long noncoding RNA (lncRNA).

To corroborate this result in the context of FLC tumors, we sought to more directly evaluate the transcriptional activity at the LINC00473 locus using our previously published chromatin run-on sequencing (ChRO-seq) data [53]. ChRO-seq enables the genome-wide detection of nascent transcription [59]. In FLC tumors, we observed very high levels of transcription on the minus strand within the ~49kb and ~63kb windows that mark the gene bodies of LINC00473 variants 1 and 2, respectively (S1A Fig). Importantly, transcriptional activity was markedly diminished in the ~250kb gap between the annotated LINC00473 locus and the RefSeq annotation for *PDE10A* (S1A Fig). These data confirm that LINC00473 transcription is separate from *PDE10A* and that it is a lncRNA that merits further analysis in FLC.

### RNA-seq on an expanded number of primary patient samples reveal LINC00473 as a top upregulated gene in FLC

To examine differentially expressed genes in FLC, we performed RNA-seq on an expanded set (Cohort 1) of FLC patient tumors (n = 35) and matched non-malignant liver (NML, n = 10)

from the Fibrolamellar Cancer Foundation biobank, and confirmed *DNAJB1-PRKACA* (DP fusion) expression in all tumor samples. Our analysis determined that LINC00473 is significantly elevated in FLC compared to NML (fold-change > 25, Fig 1A), though there is some variance of expression values across patients (Fig 1B).

We then restricted the analysis to only patient-matched FLC and NML samples in an independent cohort from the Fred Hutchinson Cancer Center (Fig 1C, n = 5), as well as in a subset of Cohort 1 (Fig 1D, n = 9; S1B Fig) and confirmed that LINC00473 is indeed significantly elevated in FLC (fold-change > 50). This finding was further validated by RT-qPCR for both isoforms of LINC00473 (TV1, TV2) in a subset of patient-matched samples (Fig 1E, n = 7; S1C Fig). A direct comparison of DP fusion and LINC00473 expression levels in FLC tumors indicated a highly significant correlation between these genes (Fig 1F, n = 12).

We then assayed several patient-derived models of FLC, notably patient-derived xenograft (PDX) tissue as well as an FLC cell line established from this PDX, and found that DP fusion and LINC00473 are concordantly elevated (Fig 1G). We also observed dramatic upregulation of both TV1 and TV2 isoforms of LINC00473 in HepG2 cells (Fig 1H) and HEK293 cells (S1C Fig) in which the ~400kb heterozygous deletion was induced by CRISPR-Cas9 to generate the DP fusion (we refer to these as HepG2-DP and HEK293-DP, respectively). Together, these data demonstrate that LINC00473 is consistently and robustly elevated in primary tumor tissue and multiple models of FLC.

## LINC00473 is enriched in FLC tumor epithelial cells

Despite the clear upregulation of LINC00473 in FLC revealed by bulk tissue analysis, it is unknown what cell type(s) within the tumor drive this signal. To address this knowledge gap, we first analyzed our recently published single-nucleus ATAC-seq data on one matched set of FLC primary tumor, metastatic tumor, and NML tissue [60]. By examining open chromatin signal on a genome-wide scale, we could infer active transcription at gene loci with single nucleus resolution. Our analysis revealed eight distinct clusters that were subsequently assigned specific cell types using established marker genes of liver-resident cells (S2A Fig). Next, we focused on the gene activity at the LINC00473 locus and determined that it is essentially restricted to FLC tumor epithelial cells (S2B and S2D Fig). Further, we examined the cellular localization pattern of LINC00473 by performing subcellular fractionation followed by RT-qPCR and found that the RNA is enriched in the nucleus, with modest cytoplasmic localization, in FLC cells (S2E Fig).

## Silencing of the DP fusion leads to LINC00473 suppression

Although it has been shown that LINC00473 expression is sensitive to PKA activity in HEK293-DP cells, this has not been investigated in the context of patient-derived FLC cells [52]. To test this, we performed siRNA-mediated knockdown of PRKACA in FLC cells generated from PDX tumors and confirmed significant suppression of both isoforms of LINC00473, as well as the DP fusion (Fig 2A).

Next, we sought to examine if LINC00473 is also suppressed by the knockdown of the DP fusion specifically. To address this question, we designed three N-acetylgalactosamine (GalNAc)-conjugated siRNAs targeting the fusion junction of *DNAJB1-PRKACA* to specifically suppress the fusion oncogene. We also designed a luciferase-targeting siRNA (siLuc) as negative control. Upon Lipofectamine-mediated transfection of the three DP fusion-targeted siRNAs (siDP#1–3) in FLC cells, we observed robust suppression of the DP oncogene and both isoforms of LINC00473, with the strongest gene silencing mediated by siDP#2 (Fig 2B).

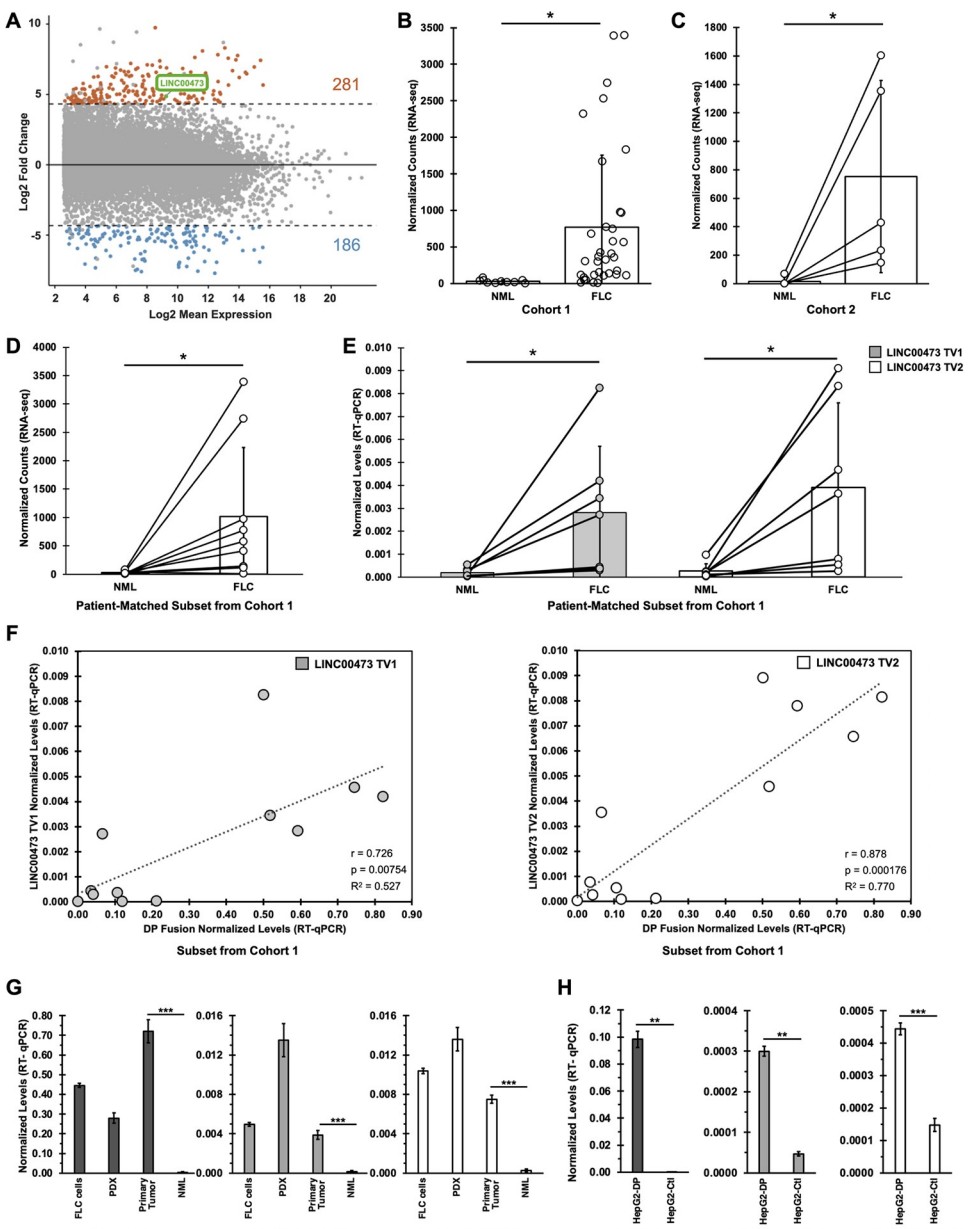

**Fig 1. RNA-seq on an expanded number of primary patient samples reveals LINC00473 as a significantly upregulated gene in FLC. (A)** MA-plot of genes that are significantly differentially expressed in primary FLC tumors (n = 35) versus non-malignant liver (NML) (n = 10). Genes filtered for expression with base mean > 1000, log2FC > 1 or < -1, and padj < 0.05 (DESeq). Dashed lines represent expression log2 FC of > 4 or < -4. Each dot represents a gene, and significant (padj < 0.05) up- and down-regulated genes are colored orange or blue, respectively. Samples comprise Cohort 1 provided by the Fibrolamellar Cancer Foundation biobank. **(B)** Normalized counts of LINC00473 as determined by RNA-seq using samples in Cohort 1 (10 NML, 35 FLC). Each dot represents a patient sample. **(C)** Normalized counts of LINC00473 expression as determined by RNA-seq using an independent set of patient-matched NML and FLC samples in Cohort 2 (5 NML, 5 FLC). Each dot is a patient sample, and matched NML/FLC samples are indicated with a connecting line. Samples comprise Cohort 2 provided by the Fred Hutchinson Cancer Center. **(D)** Normalized counts of LINC00473 in FLC patient tumors and matched NML in a subset of samples from Cohort 1. **(E)** Normalized levels of LINC00473 TV1 and TV2 as determined by RT-qPCR in patient-matched NML and FLC from Cohort 1 (n = 7 per group). **(F)** Pearson correlation between transcript levels of DP fusion and LINC00473 isoforms TV1 (left) and TV2 (right) in a subset of primary FLC tumors from Cohort 1 (n = 12). Dashed lines represent the fitted linear regression curve. Each dot represents a FLC patient tumor, and identical tumors were assessed for both TV1 and TV2. $R^2$ and p values are indicated. **(G, H)** Normalized levels of DP fusion transcript and LINC00473 isoforms TV1 and TV2 RNA in multiple models of FLC and matched NML (G), and HepG2-DP cells relative to non-targeting

sgRNA control (HepG2-Ctl) (H). n = 3 biological replicates for PDX, primary tumor and NML, and n = 3 technical replicates for FLC cells, HepG2-DP and HepG2-Ctl cells. For gene expression via RT-qPCR in E, F, G, and H, normalized levels represent 2^(-dCt) values using *RPS9* for normalization. Data are represented as mean ± SD. P values are calculated by 2-tailed Student's t-test. *p < 0.05, **p < 0.01, ***p < 0.001.

Lipid-based reagents for transfection can damage or influence cells in unpredictable ways and are also not tractable for in vivo delivery. To circumvent this issue, we sought to evaluate gene silencing in FLC cells through transfection-free uptake of siRNAs. We tested several siRNA delivery systems, including encapsulation in lipid nanoparticles (LNPs) or conjugation with GalNAc, 2'-O-hexadecyl (C16), or cholesterol. As a proof-of-concept, we targeted *CTNNB1*, which is expressed in FLC cells, using a siRNA sequence previously shown to drive potent and specific gene knockdown. We discovered that, among the methods tested, LNP encapsulation was the most effective delivery method in FLC cells (Fig 2C and 2D). Notably, we found >90% reduction in gene expression at concentrations as low as 12.5nM. Interestingly, treatments at 5- and 25-fold higher concentrations did not improve gene silencing efficacy, suggesting that a concentration in the range of 2.5nM—12.5nM is likely sufficient for gene knockdown (Figs 2D and S3A–S3D). Together, these data demonstrated that siDP#2 (and to a slightly lesser extent, siDP#1) exhibited potent DP fusion silencing and that LNP encapsulation yielded efficient siRNA free uptake in FLC cells.

We next tested whether LNP-mediated delivery of siDP#1 or siDP#2 could achieve potent and specific knockdown of DP fusion in FLC cells. We encapsulated siDP#1, siDP#2, or siLuc control in LNPs and subsequently treated FLC cells at 5nM. Subsequent western blots were performed using an antibody that binds to the carboxyl terminus of wild-type PRKACA (WT PKA) and detects both native PKA and DP fusion protein. Consistent with previous reports, the major form of the DP fusion was detected at 48kDa [11]. In contrast, the larger band at 51kDa corresponds to the minority oncoprotein, which has been shown previously to be present in the PDX model used to derive the FLC cell line [11,61]. We observed a significant reduction in DP fusion protein expression in response to siDP-LNPs (Fig 2E and 2F). Although we also observed decreased WT PKA protein abundance, the suppression of DP fusion protein was ~3-fold greater by comparison (Fig 2F). WT DNAJB1 levels were unaltered after treatment (Figs 2F and S3E). Importantly, we found that knockdown of the DP fusion by siDP-LNP led to a marked downregulation of LINC00473 and other FLC marker genes (Figs 2G and S3F). Together, we demonstrate that a siRNA-LNP approach leads to potent silencing of the DP fusion at the RNA and protein level and robust LINC00473 suppression. These observations provide strong evidence that LINC00473 expression is driven by the DP fusion in FLC.

## FLC cell growth is altered following LINC00473 knockdown or overexpression

To interrogate the function of LINC00473 in FLC, we developed complementary gene knockdown and overexpression approaches using lentiviral integration of short hairpin RNAs (shRNAs) or cDNA to enable stable gene modulation in FLC cells. For gene knockdown, two distinct LINC00473-targeting shRNAs (sh473-2, sh473-4) that were previously reported to strongly downregulate LINC00473 TV1 and non-targeting control (shCtl) were cloned into lentiviral plasmids and subsequently transduced in FLC cells [47]. Following antibiotic selection, the majority of transduced FLC cells were predicted to express a single copy of shRNA to mitigate off-target effects. Using RT-qPCR to query gene silencing, we showed that sh473-2 and sh473-4 yield efficient gene knockdown by 80% and 50%, respectively (Fig 3A). Interestingly, although the shRNAs were originally designed to target LINC00473 TV1, the expression

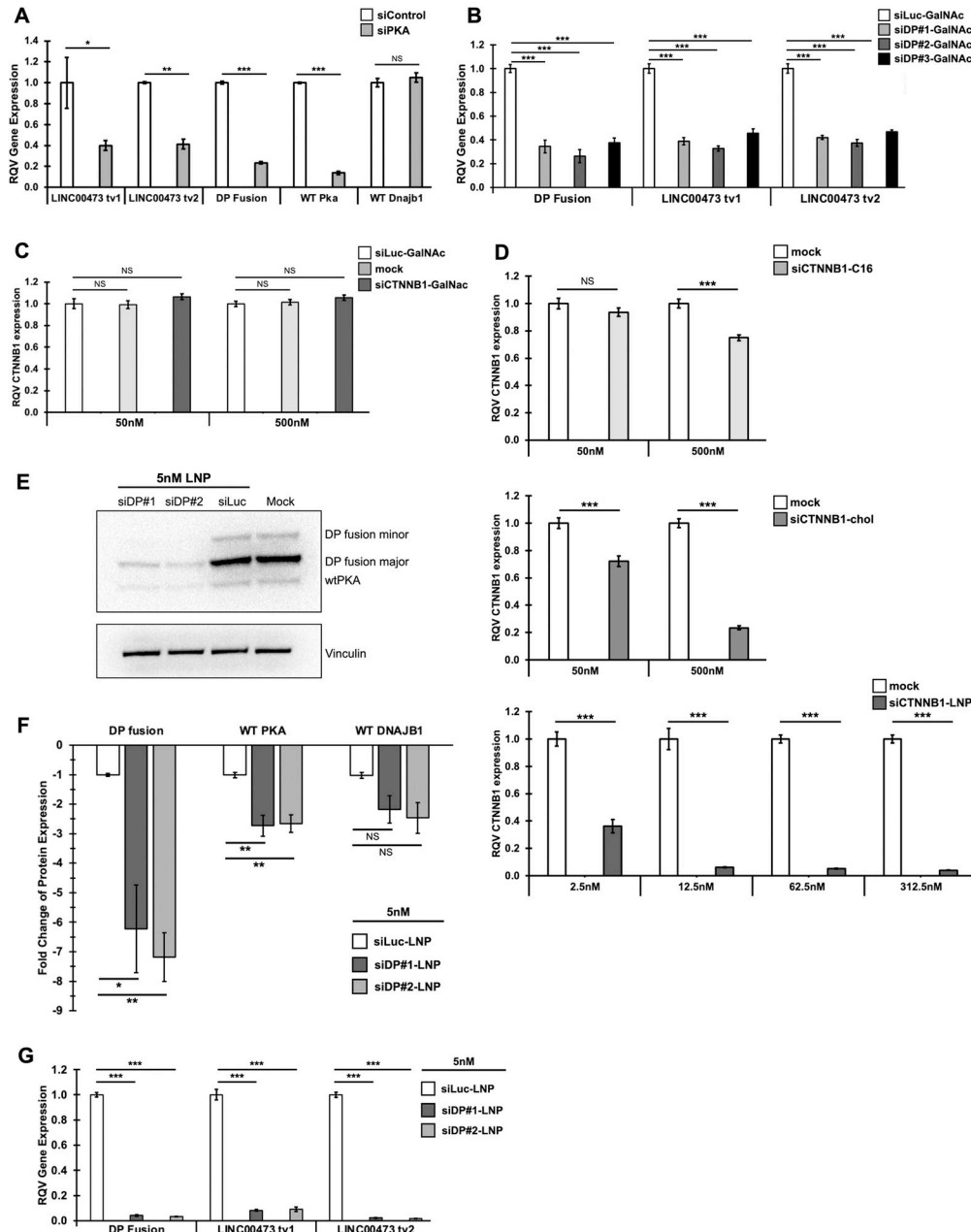

**Fig 2. Silencing of the DP fusion leads to the suppression of LINC00473. (A)** Gene expression following transfection of siPKA or non-targeting siRNA negative control (siControl) at 48nM over 72 hours (n = 2 biological replicates)**. (B)** Gene expression from RT-qPCR following lipofectamine-mediated transfection of three independent N-acetylgalactosamine (GalNAc)-conjugated siRNAs targeting the DP fusion (siDP#1-GalNAc, siDP#2-GalNAc, siDP3-GalNAc) and siLuciferase (siLuc-GalNAc) negative control at 50nM over 48 hours. (n = 3)**. (C)** *CTNNB1* gene expression from RT-qPCR following free uptake of siCTNNB1-GalNAc, siLuciferase (siLuc-GalNAc) negative control at 50nM and 500nM, or mock negative control, over 48 hours (n = 3)**. (D)** *CTNNB1* gene expression from RT-qPCR following free uptake of siCTNNB1 conjugated to 2′-O-hexadecyl (C16) or cholesterol (chol), at 50nM or 500nM and mock negative control, or siRNA encapsulated in lipid nanoparticles (LNP) at 2.5nM, 12.5nM, 62.5nM, and 312.5nM, or mock negative control, over 48 hours (n = 3)**. (E)** Representative immunoblot of protein expression of DNAJB1-PRKACA (DP) fusion. DP fusion is detected with a protein kinase A catalytic α subunit (PKA) antibody. WT PKAc, DP fusion major, and DP fusion minor are identified. Lane 1, siDP#1-LNP; Lane 2, siDP#2-LNP; Lane 3, siLuciferase (siLuc-LNP) negative control; Lane 4, mock negative control. siRNA-LNP treatments at 5nM, or mock condition, over 96 hours. Following 5nM treatment with siRNA-LNPs or mock condition over 96 hours. Vinculin loading control is shown in the lower panel and run on the same blot (n = 3)**. (F)** Fold change of protein levels in Fig 3E, relative to siLuc negative control (n = 3)**. (G)** Gene expression from RT-qPCR following free uptake of

siDP#1-LNP, siDP#2-LNP, and siLuc-LNP at 5nM treatment over 96 hours in FLC cells (n = 3). Relative Quantitative Values represent the change of any normalized measurement relative to the control group. Data are represented as mean ± SD. P values are calculated by 2-tailed Student's t-test. *p < 0.05, **p < 0.01, ***p < 0.001.

of the second isoform was also modestly decreased upon treatment (Fig 3B). For gene upregulation, the cDNA sequence of LINC00473 TV1 was cloned into a LeGO-lnc lentiviral vector (LeGO-473ox). We followed a similar method of transduction as with the shRNA integration, resulting in a dramatic 85-fold increase in gene expression levels compared to empty LeGO-lnc vector control (LeGO-Ctl) (Figs 3C, S4A).

We first assessed the effect of LINC00473 on FLC cell growth using our polyclonal knockdown and overexpression cell lines. By examining cell counts over five days, we found that the stable downregulation of LINC00473 using sh473-2 or sh473-4 significantly decreased cell growth across two independent trials (Fig 3D). Concordantly, increased cell growth was observed in polyclonal FLC cells expressing elevated levels of LINC00473 relative to LeGO-Ctl (Fig 3E). To mitigate the effects of genetic drift in cell culture, we performed limiting dilutions on our engineered polyclonal cells to develop several independent monoclonal cell lines. As expected, monoclonal cells displayed a range of LINC00473 downregulation and overexpression as determined by RT-qPCR (S4B and S4C Fig). To confirm that LINC00473 regulates FLC cell growth, we selected three monoclones from each group for cell growth quantification: sh473-2 (clones A3, A4, A6), sh473-4 (clones B2, B3, B4), shCtl (clones C2, C5 C6), LeGO-473ox (clones D3, D4, D6), and LeGO-Ctl (clones E1, E5, E6). In line with our prior data, monoclonal cell growth decreased under conditions of LINC00473 suppression (Fig 3F), which is validated in similarly engineered HEK293-DP monoclones with shRNA integration (S4D–S4F Fig) (Fig 3G), and increased levels of LINC00473 led to enhanced cell growth (Fig 3G).

To further validate the role of LINC00473 in regulating cell growth in FLC, we next examined the effects of LINC00473 inhibition and upregulation on anchorage-independent colony formation. Monoclonal cell lines harboring LINC00473 knockdown (clone A3) or overexpression (clone D3), as well as the corresponding controls (clones C2 and E6, respectively), were analyzed for colony growth at 14 days after seeding. Consistent with our cell growth data, FLC cells exhibited drastically decreased colony expansion upon LINC00473 silencing (Fig 3H), and elevated levels of LINC00473 resulted in significantly greater colony growth (Fig 3I). Taken together, these data confirm that LINC00473 is a key regulator of cell growth in FLC.

## LINC00473 increases FLC cell survival by suppressing apoptosis

To determine which genes and pathways are most altered in response to perturbation of LINC00473, we performed RNA-seq on multiple independent monoclonal cell lines with LINC00473 overexpression (LeGO-473ox clones D3, D4) or empty vector control (LeGO-Ctl clones E5, E6). Our differential gene expression analysis identified 1403 upregulated genes and 1374 downregulated genes in LeGO-473ox cells compared to LeGO-Ctl controls. As expected, we found that LINC00473 itself is one of the top-most elevated genes in LeGO-473ox (Fig 4A). Principal component analysis (PCA) showed stratification of samples by condition based on gene expression profiles (Fig 4B). Furthermore, unsupervised hierarchical clustering analysis confirmed that LeGO-473ox clones segregated completely from LeGO-Ctl control clones (Fig 4C).

We proceeded to evaluate the expression of key FLC marker genes known to be highly elevated in primary tumors, including *CA12*, *VCAN*, *OAT*, *TESC*, and *RPS6KA2*. We confirmed that the RNA levels of the majority of the queried genes were robustly increased in FLC cells in which LINC00473 was overexpressed (Figs 4D and S5A), suggesting that LINC00473 may play a role in regulating the expression of key FLC-associated genes.

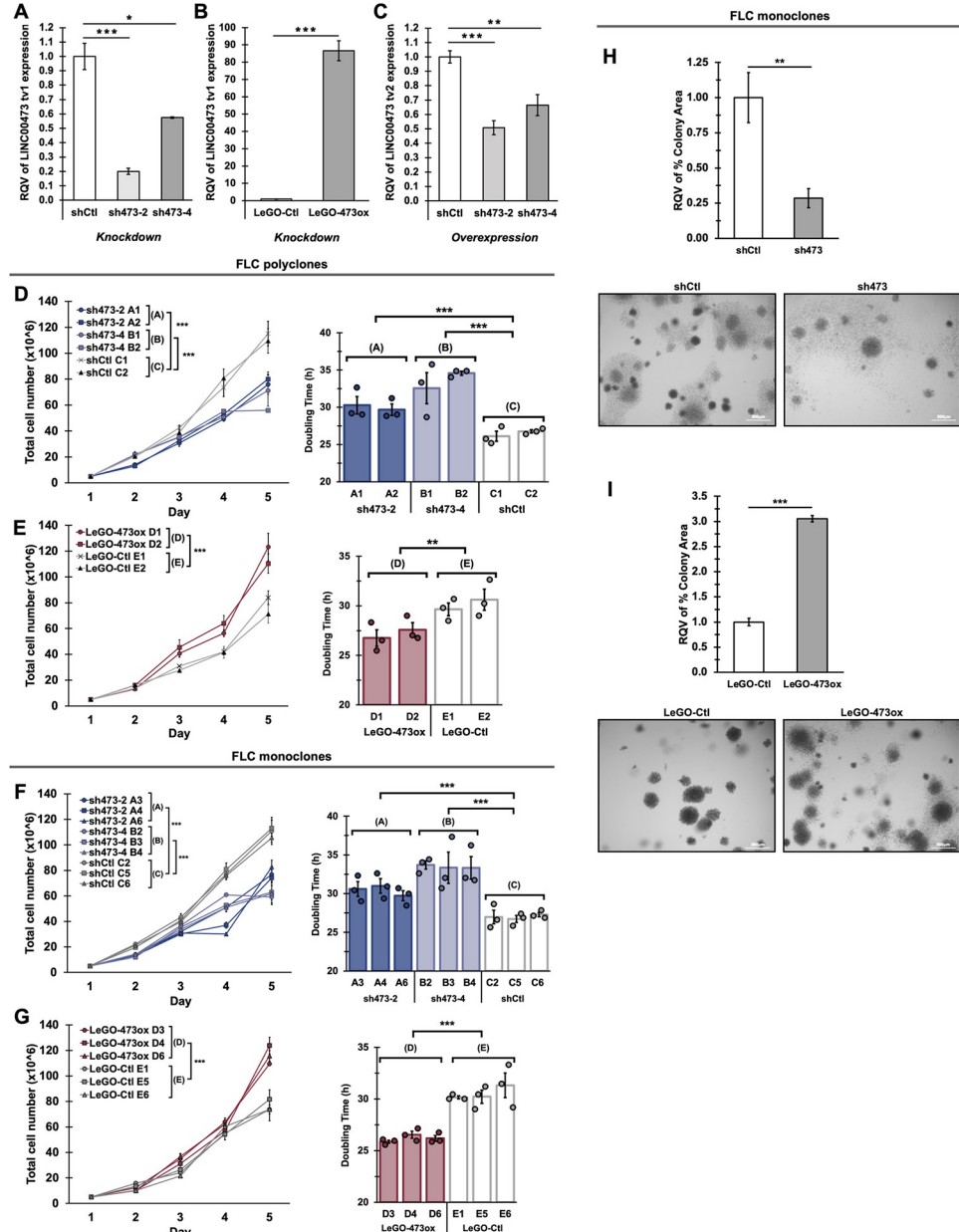

**Fig 3. FLC cell growth is altered following LINC00473 knockdown and overexpression. (A, B)** LINC00473 TV1 expression from RT-qPCR in FLC polyclonal cells following lentiviral transfection enabling stable gene knockdown using two independent shRNAs (sh473-2, sh473-4) or non-targeting shRNA control (shCtl) (A) or overexpression using cDNA plasmid encoding LINC00473 TV1 (LeGO-473ox) or empty-vector control (LeGO-Ctl) (B). **(C)** Expression of LINC00473 TV2 isoform in FLC polyclonal cells following lentiviral transfection enabling stable gene knockdown using two independent shRNAs (sh473-2, sh473-4) or non-targeting shRNA control (shCtl). **(D, E)** Cell growth curve (left) and doubling time in hours (right) of FLC polyclonal cells with stable LINC00473 knockdown (sh473-2, sh473-4) and non-targeting control (shCtl) (D) or overexpression using LINC00473 cDNA (LeGO-473ox) and empty vector control (LeGO-Ctl) (E). Each polyclonal cell line was quantified 2 times across independent passages (A1, B1, C1 from passage 1; A2, B2, C2 from passage 2), and each point is the average total cell count of 3 replicates. **(F, G)** Cell growth curve (left) and doubling time in hours (right) of FLC monoclonal cells with stable LINC00473 knockdown (sh473-2: clones A3, A4, A6; sh473-4: clones B2, B3, B4) and non-targeting control (shCtl: clones C2, C5, C6) (F), or overexpression (LeGO-473ox: clones D3, D4, D6) and empty vector control (LeGO-Ctl: clones E1, E5, E6) (G). Each monoclonal cell line was quantified 2 times across independent passages, and each point is the average total cell count of 3 replicates. **(H, I)** Anchor-age-independent soft agar colony formation of FLC monoclonal cells expressing stable LINC00473 knockdown (sh473) relative to control (shCtl) (H), and overexpression (LeGO-473ox) relative to control (LeGO-Ctl) (I). Representative images of nitro blue tetrazolium-stained colonies are shown 14 days

after seeding from A3, C2, D3 and E6 (2 trials, n = 3 per monoclonal cell line). Relative Quantitative Values represent the change of any normalized measurement relative to the control group. Scale bars: 500μM. Bar chart data are represented as mean ± SD, n = 3, unless otherwise stated. Statistical testing on cell growth curves was performed using a 2-tailed Student's t-test following area under the curve analysis using the definite integral of each fitted polynomial function. Otherwise, P values are calculated by 2-tailed Student's t-test. *$p < 0.05$, **$p < 0.01$, ***$p < 0.001$.

Next, we performed pathway enrichment analysis using Enrichr [62]. Strikingly, genes upregulated by LINC00473 were significantly enriched in metabolic pathways, including glycolysis (MSigDB, Fig 4E), pentose phosphate pathway, galactose metabolism, and mitochondrial respiratory chain dysfunction (S5B Fig). Additionally, cancer metabolism-related genes, including *PFKM*, *IDH3G*, *PYCR1*, and *BCAT1*, were strongly elevated in expression in response to increased LINC00473 levels (Fig 4E). We also found that genes downregulated by LINC00473 were significantly enriched in apoptosis and p53 pathways (MSigDB, Figs 4F and S5C). Notably, genes that encode for pro-apoptotic proteins, such as *BCL2*, *TRIB3*, *KLF10*, and *GADD45B*, were markedly suppressed in the context of LINC00473 upregulation (Fig 4F). Together, these data implicate LINC00473 as a regulator of metabolism and a suppressor of apoptotic pathways in FLC cells.

To investigate whether LINC00473 increases FLC cell growth by altering the rate of cell proliferation, cell death, or both, we performed EdU incorporation assays and TUNEL assays in FLC monoclones harboring sh473-mediated LINC00473 knockdown or shCtl and LeGO-473ox-driven LINC00473 overexpression or LeGO-Ctl. EdU incorporation rates did not indicate notable differences in cell proliferation across groups (Fig 4G). In contrast, TUNEL staining revealed a striking reduction of cell death in the context of LINC00473 over-expression and a significant increase in cell death upon LINC00473 knockdown (Fig 4H). Thus, decreased apoptosis in LINC00473-overexpressing FLC cells likely underlies their elevated growth relative to control cell lines. These findings further implicate LINC00473 as a regulator of FLC cell growth that functions, in part, to suppress apoptotic pathways.

Due to the recurrent theme of metabolic pathways among the genes upregulated by LINC00473-overexpressing cells, we investigated which of these genes are also altered in FLC tumors. By comparing the genes that are significantly elevated in FLC tumors relative to NML (n = 1667) and in LeGO-473ox cells relative to control (n = 1403), we found a significant number of overlapping genes (n = 188, Fig 4I). We determined that these 188 genes are over-represented in pathways linked to proline metabolism (Fig 4I), metabolic effects of oncogenes, cancer metabolic reprogramming, glutamine in cancer metabolism, and proline metabolism (Figs 4I and S5D). Remarkably, energy metabolism-related genes such as *BCAT1*, *CTH*, *PFKM*, *PRODH*, *PYCR1*, and *SLC7A11* stood out among the overlapping genes as the most highly upregulated in both FLC tumors and in FLC cells with increased LINC00473 expression.

A parallel analysis was performed on the genes most significantly downregulated in both FLC tumors and LeGO-473ox cells. This examination revealed an intersection of 91 genes, including *FOS*, *GADD45A/B*, *KLF10/11*, and *TRIB3*, which are involved in p53 signaling and apoptosis (Figs 4J and S5E). Taken together, these findings strongly suggest that LINC00473 plays a role in dysregulated energetics in FLC and the suppression of apoptosis in FLC.

## LINC00473 functions to increase glycolysis and mitochondrial activity in FLC cancer metabolism

We next explored how LINC00473 regulates cellular energetics in FLC. To investigate this, we performed Seahorse metabolic analyses on FLC monoclones harboring stable LINC00473 knockdown (sh473) and non-targeting control (shCtl) or overexpression (LeGO-473ox) and empty vector control (LeGO-Ctl). This enabled the examination of glycolysis using

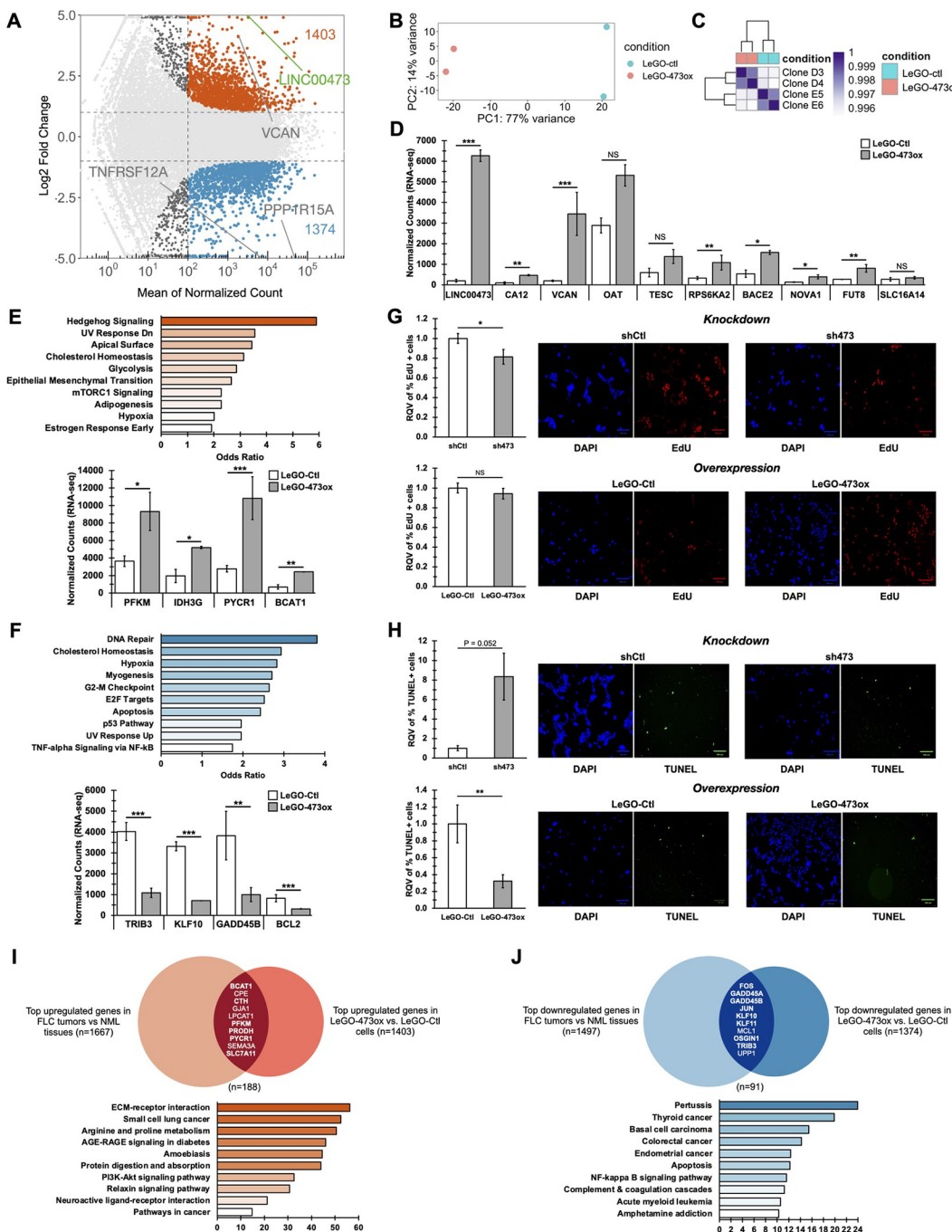

**Fig 4. LINC00473 increases FLC cell survival by suppressing apoptosis. (A)** MA plot showing differentially expressed genes in FLC cells that stably overexpress LINC00473 (LeGO-473ox) relative to empty vector control (LeGO-Ctl). Genes filtered for expression with base mean > 100, log2FC > 0.5 or <-0.5, and padj < 0.05 (DESeq). Dashed lines represent log2 FC of 1 and -1 (horizontal) and mean normalized count = 100 (vertical). Up- and down-regulated genes are colored orange or blue, respectively. **(B)** Principal component analysis of DESeq normalized rlog counts with cell treatment information shown by color. LeGO-Ctl (n = 2), and LeGO-473ox (n = 2) are shown in orange and blue, respectively. **(C)** Hierarchical clustering analysis of gene expression profiles across samples using DESeq normalized rlog counts. FLC cells with LINC00473 overexpression (LeGO-473ox) or empty vector control (LeGO-Ctl) are indicted by orange and blue boxes, respectively. **(D)** Normalized counts for specific FLC marker genes from RNA-seq from LeGO-473ox and LeGO-Ctl control FLC monoclonal cells. **(E, F)** MSigDB Hallmark pathway enrichment analysis of significantly upregulated genes (n = 1403) (A) and downregulated genes (n = 1374) (B), respectively. Normalized counts for metabolism-related genes from RNA-seq from LeGO-473ox and LeGO-Ctl control FLC monoclonal cells. **(G)** EdU incorporation in FLC cells with stable LINC00473

knockdown (sh473) compared to control (shCtl), and LINC00473 overexpression (LeGO-473ox) compared to control (LeGO-Ctl). Representative images of DAPI- and EdU-stained cells show total and proliferative cells, respectively. **(H)** TUNEL staining in FLC cells with stable LINC00473 knockdown (sh473) compared to control (shCtl), and LINC00473 overexpression (LeGO-473ox) compared to control (LeGO-Ctl). Representative images of DAPI- and TUNEL-stained cells show total and apoptotic cells, respectively. **(I)** Gene list overlap analysis using significantly upregulated genes in FLC tumors relative to NML (n = 1667), and in LeGO-473ox cells relative to control (n = 1403). Bolded genes indicate proteins related to metabolism. KEGG pathway enrichment analysis of the intersecting 188 genes. Significance of the overlap (p = 4.81x10$^{-20}$) was calculated by hypergeometric test. **(J)** Gene list overlap analysis using significantly downregulated genes in FLC tumors relative to NML (n = 1497), and in LeGO-473ox cells relative to control (n = 1374). Bolded genes indicate proteins related to apoptosis. KEGG pathway enrichment analysis of the intersecting 91 genes. Overlap of n = 91 genes was not significant following a hypergeometric test. Bar chart data in G and H are represented as mean across n = 6 ± SD. Relative Quantitative Values represent the change of any normalized measurement relative to the control group. Pathway enrichment figures indicate pathways with p-values < 0.05 and bar color intensity represents odds ratio.
Scale bars represent 100 μm. P values are calculated by 2-tailed Student's t-test. *p < 0.05, **p < 0.01, ***p < 0.001 unless otherwise indicated.

extracellular acidification rate (ECAR), which monitors changes in pH levels as a readout of lactate production. Several parameters of glucose metabolism are measured as cells respond to a series of compounds that alter glycolytic function. We observed that LINC00473 upregulation led to a marked elevation in glucose utilization, as shown by increased levels of glycolysis and glycolytic capacity in LeGO-473ox cells relative to LeGO-Ctl (Fig 5A and 5B). In contrast, LINC00473 silencing resulted in a trend toward decreased glycolysis, although the result was not significant (Fig 5C and 5D).

Next, we investigated the effect of LINC00473 on mitochondrial respiration by quantifying oxygen consumption rate (OCR). By tracking oxygen levels as cells respond to a series of modulators that perturb the electron transport chain, we were able to quantify several parameters of mitochondrial respiration in our knockdown and overexpression FLC cell lines. Spare respiration capacity (SRC) is calculated by subtracting maximum respiration from basal respiration and reveals the adaptability of mitochondria to meet acute energy demands, thereby serving as an indicator of mitochondrial fitness in stress conditions. Importantly, SRC dysregulation is a known cancer cell phenotype, and we have observed that SRC is dramatically depleted in FLC cells (personal communication, Donald Long, Jr.). In the context of LINC00473 overexpression, we observed a modest elevation in basal respiration in FLC cells compared to empty vector control (Fig 5E and 5F). Knockdown of LINC00473 led to a prominent rescue of SRC (Fig 5G and 5H). In sum, these data suggest that LINC00473 may increase baseline mitochondrial activity at the cost of reduced mitochondrial fitness in FLC cells.

## LINC00473 promotes FLC tumor growth

To investigate if the growth-promoting effect of LINC00473 observed in FLC cells persists in the *in vivo* context, we studied tumor growth progression in a xenograft animal model. We implanted FLC cells with LINC00473 overexpression (LeGO-473ox) or empty vector control (LeGO-Ctl) into mice by subcutaneous injection and quantified relative volume progression in each tumor for eight days. In line with our *in vitro* data, we observed that LINC00473 induction significantly increases tumor growth rate (Fig 6A and 6B). Further, we assessed gene expression levels in LINC00473-overexpression tumors and confirmed strong upregulation of LINC00473, as well as unaltered expression of the DP fusion, relative to tumor controls (Figs 6C and S6A).

We next explored the pathways LINC00473 activates *in vivo*. To examine the gene expression programs in tumors generated from LeGO-473ox cell implantation, we performed RNA-seq on a subset of tumor samples (n = 4 per group). Our differential gene expression analysis identified 953 upregulated genes and 1468 downregulated genes in LeGO-473ox cells

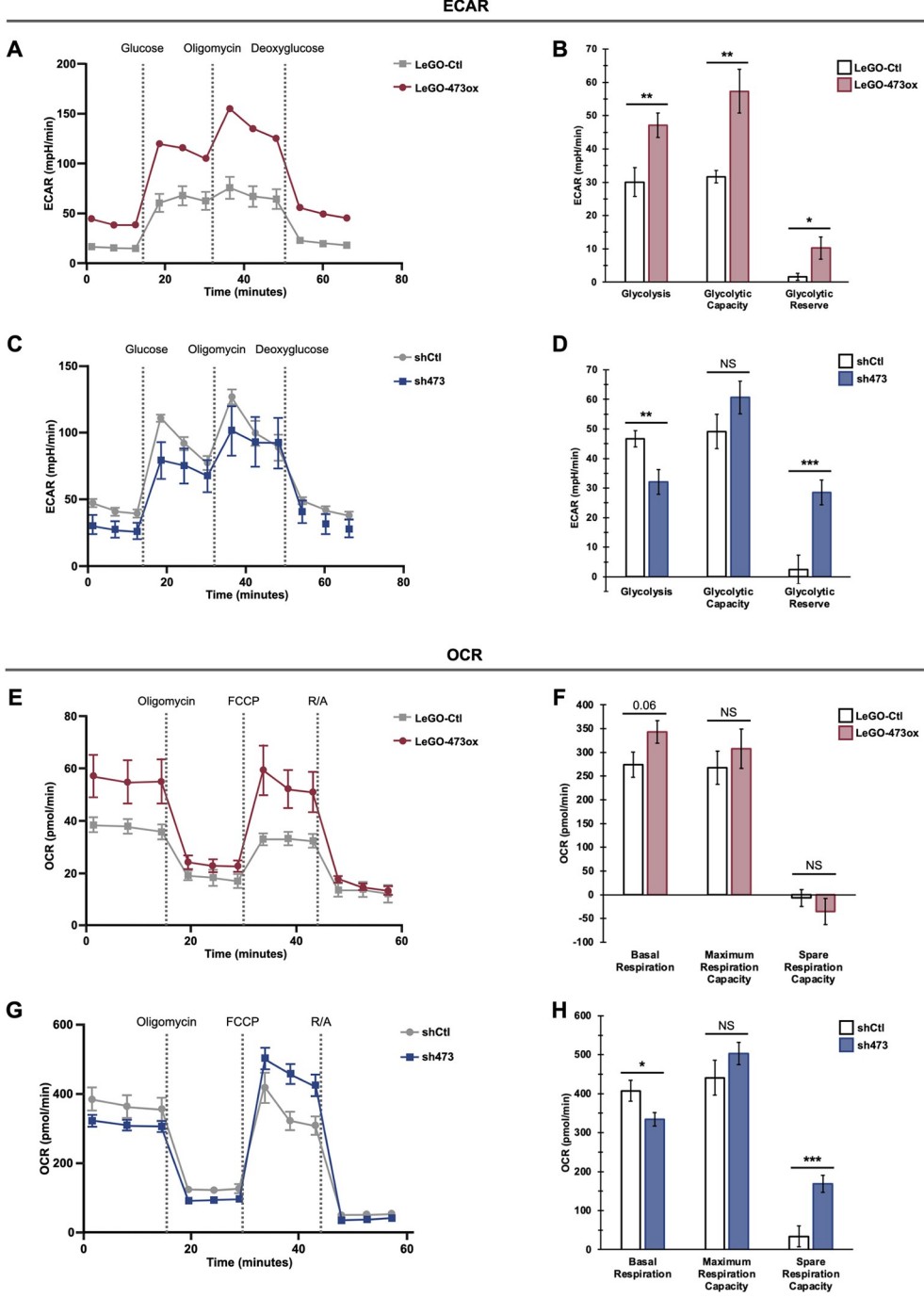

**Fig 5. LINC00473 increases glycolysis and alters mitochondrial activity in FLC.** (**A**) Representative graph of extracellular acidification rate (ECAR) of FLC cells with stable LINC00473 overexpression (LeGO-473ox) and empty vector control (LeGO-Ctl). (**B**) Quantification of glycolysis, glycolytic capacity and glycolytic reserve in LINC00473-overexpression (LeGO-473ox) FLC cells compared to empty-vector control (LeGO-Ctl). (**C**) Representative graph of ECAR of FLC cells with stable LINC00473 knockdown (sh473) and non-targeting control (shCtl). (**D**) Quantification of glycolysis, glycolytic capacity and glycolytic reserve in LINC00473-knockdown (sh473) FLC cells compared to non-targeting control (shCtl). Data are representative of 2 or 3 independent trials, respectively, of n = 5 replicates per experiment. € Representative graph of oxygen consumption rate (OCR) of FLC cells with overexpression (LeGO-473ox) and control (LeGO-Ctl). (**F**) Quantification of basal respiration, maximum respiration capacity, and spare respiratory capacity in LINC00473-overexpression (LeGO-473ox) FLC cells compared to empty-vector control (LeGO-Ctl). (**G**) Representative graph of OCR of FLC cells with stable LINC00473 knockdown (sh473)

and non-targeting control (shCtl). **(H)** Quantification of basal respiration, maximum respiration capacity, and spare respiratory capacity in LINC00473-knockdown (sh473) FLC cells compared to non-targeting control (LeGO-Ctl). Data are representative as mean ± SD across 2 or 3 independent trials, respectively, of n = 5 replicates per experiment. P values are calculated by 2-tailed Student's t-test. *p < 0.05, **p < 0.01, ***p < 0.001.

compared to LeGO-Ctl controls (Fig 6D). As expected, principal component analysis stratified samples by condition based on gene expression profiles (Fig 6E). Further, we assessed pathway enrichment using Enrichr and determined that oxidative phosphorylation is strikingly activated by LINC00473 in cell-derived xenograft tumors (Fig 6F). Interestingly, hedgehog and mTOR signaling pathways are similarly upregulated in response to LINC00473 expression in both *in vitro* (Fig 4E) and *in vivo* contexts (Fig 6F). A parallel analysis was performed on the pathways most significantly suppressed by LINC00473, including DNA repair in LeGO-473ox cells (Fig 4F) and primary tumors (Fig 6G). Notably, TNF-alpha signaling via NF-kB and cholesterol homeostasis are robustly suppressed by LINC00473 in LeGO-473ox tumors (Fig 6G) and primary FLC tumors (S1E).

Taken together, these results confirm the role of LINC00473 as a regulator of FLC growth *in vivo*. A working model of the role of LINC00473 in FLC is shown in Fig 7.

## Discussion

FLC is a rare and aggressive type of liver cancer that often presents at advanced stages owing in part to the lack of diagnostic biomarkers and known risk factors. Faced with a paucity of treatment options, FLC patients suffer from low survival rates. Over the past decade, there has been exciting progress in pursuing driver genes, most notably in identifying the signature *DNAJB1-PRKACA* (DP) fusion oncogene. However, the development of specific inhibitors of the DP fusion remains a challenge, and an urgent and unmet need for effective therapies persists [17,18]. The mechanisms by which the fusion oncogene transforms cells remain poorly understood. To gain deeper insight into FLC molecular etiology, we leveraged multiple genome-scale strategies and functional approaches, which revealed LINC00473 as a critical regulator of FLC phenotypes.

LINC00473 is a primate-specific long noncoding RNA, and its normal functions are poorly understood [55,58]. Earlier reports provided little to no support for coding potential at the LINC00473 locus [47]. Pruunsild et al. recently showed that the open reading frame could be translated; however, for various reasons, the authors concluded that the function of the putative protein is likely negligible compared to the lncRNA transcript [55]. In the future, it will be interesting to evaluate whether any small peptides are produced from this locus in FLC tumors.

LINC00473 has been reported in different subcellular compartments in distinct cellular contexts and disease states [55–57]. For instance, in non-small cell lung tumors, Chen et al. found that LINC00473 was primarily nuclear with modest cytoplasmic expression. In contrast, Tran et al. showed that LINC00473 can localize to the mitochondria in adipocytes [47,58]. Known protein binding partners of LINC00473 are NONO in several cell types as well as PLIN1 in adipocytes [47,48,58]. In normal conditions, LINC00473 exhibits a very restricted expression pattern across tissues, with the greatest abundance in the ovary, pituitary gland, and fallopian tube [63].

The limited literature on LINC00473 has impeded understanding of its biological role and molecular regulation in disease. However, LINC00473 has been reported to be elevated in several cancer types and described as a predictor of poor prognosis [21,47–50]. For instance, one study identified LINC00473 as a cAMP/CREB target gene required for the growth and survival

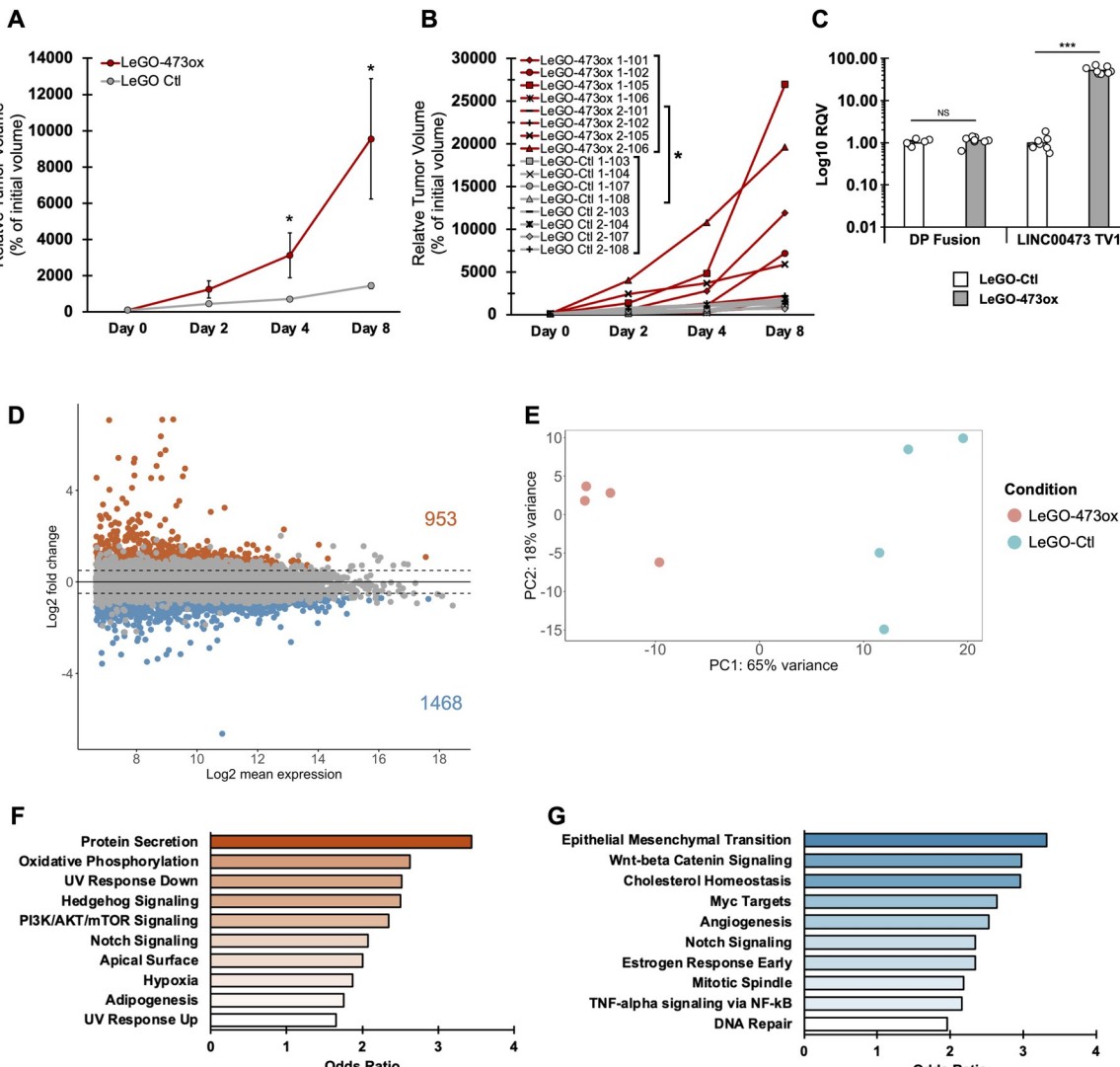

**Fig 6. LINC00473 is a regulator of FLC tumor growth in vivo. (A)** Growth curves of subcutaneous tumors following transplantation of FLC monoclonal cells into the flank of NOD–Scid–Il2rg-deficient (NSG) mice. Tumors formed from transplantation of FLC monoclonal cells with stable LINC00473 overexpression (LeGO-473ox) and empty vector control (LeGO-Ctl). Relative tumor volume is determined as the percent growth compared to initial volume at Day 0 when tumors were first detectable and measured with calipers. n = 8 per group (4 female, 4 male). Data are represented as mean ± SD. **(B)** Individual growth curves of each subcutaneous tumor in (A). n = 4 mice per trial; 2 independent trials were conducted. **(C)** Gene expression of DP fusion transcript and LINC00473 from RT-qPCR in LINC00473-overexpression (LeGO-473ox) and empty-vector control (LeGO-Ctl) subcutaneous FLC tumors. **(D)** MA plot showing differentially expressed genes in LINC00473 overexpressing tumors (LeGO-473ox) relative to empty vector control tumors (LeGO-Ctl). Genes filtered for expression with base mean > 100, log2FC > 0.5 or <-0.5, and padj < 0.05 (DESeq). Dashed lines represent log2 FC of 1 and -1 (horizontal) and mean normalized count = 100 (vertical). Up- and down-regulated genes are colored orange or blue, respectively. **(E)** Principal component analysis of DESeq normalized rlog counts with tumor type information shown by color. LeGO-Ctl (n = 4 tumors), and LeGO-473ox (n = 4 tumors) are shown in orange and blue, respectively. **(F, G)** MSigDB pathway enrichment analyses on the top upregulated genes (n = 953) (G) or downregulated genes (n = 1468) (H) in LeGO-473ox vs LeGO-Ctl tumors. Pathways with p values <0.05 are shown and bar color intensity represents odds ratio. Relative Quantitative Values represent the change of any normalized measurement relative to the control group. Statistical testing on tumor growth was performed using a 2-tailed Student's t-test following area under the curve analysis using the definite integral of each fitted polynomial function. Otherwise, P values are calculated by 2-tailed Student's t-test. *p < 0.05, **p < 0.01, ***p < 0.001.

of non-small cell lung tumors with an inactivating mutation in the tumor suppressor gene *LKB1* [47]. Other reports demonstrated that LINC00473 is downstream of the fusion oncoprotein CRTC1-MAML2 in mucoepidermoid carcinoma, and that the deletion of CRTC2/3 leads

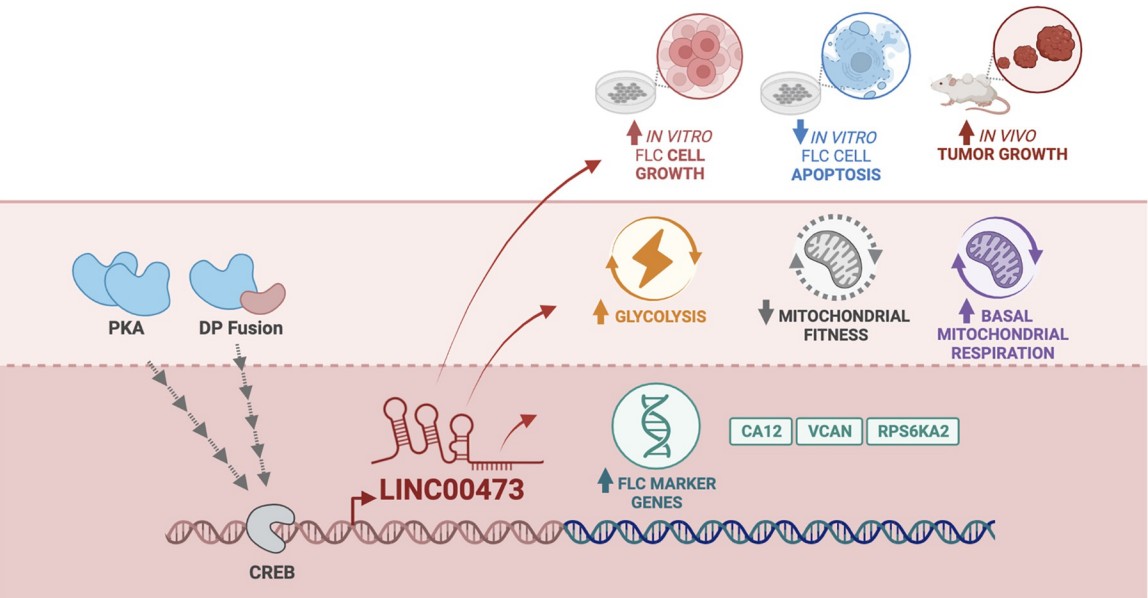

**Fig 7. Model of LINC00473 function in FLC.** Summary of the upstream regulators of LINC00473 and its role in regulating FLC-marker gene expression, glycolysis and mitochondrial energetics, and cancer cell expansion *in vitro* and *in vivo* to promote tumor growth in fibrolamellar carcinoma. Black arrows are directly supported by our study. CREB denotes camp-responsive element binding protein, and PKA denotes protein kinase A. Schematic was created with BioRender.com.

to decreased levels of LINC00473 [48,64]. In contrast, LINC00473 was discovered to be transcriptionally silenced due to promoter hypermethylation in colorectal cancer and was suppressed in osteosarcoma cells by the direct binding of ZBTB7A (encoding Pokemon) to the LINC00473 promoter [50,65].

LINC00473 has also been implicated as a target of cAMP/CREB signaling beyond malignant contexts. For instance, Liang et al. uncovered that LINC00473 mediates decidualization of human endometrial cells in response to cAMP signaling [54]. Others found that the ectopic expression of LINC00473 in mouse neurons leads to the upregulation of CREB target genes *Bdnf* and *Rheb*, which participate in a positive feedback loop to maintain CREB signaling [55]. In this study, we determined that in FLC, LINC00473 is responsive to the inhibition of the DP fusion and wild-type PKA. Also, our previous analyses revealed that a subset of cholangiocarcinoma tumors characterized by the *ATP1B1-PRKACA* fusion, which retains the same *PRKACA* exons as the DP fusion, express elevated levels of LINC00473 [53]. The strong connection between aberrant PKA signaling and LINC00473 expression is likely mediated by CREB transcription factor activity. Our group previously reported that the LINC00473 locus is associated with an FLC tumor-specific super-enhancer enriched for CREB-binding motifs. We demonstrate in this study that its overexpression leads to the upregulation of CREB target genes [53]. Taken together, the findings point to an important role for aberrant PKA signaling, including the DP fusion, in driving LINC00473 expression mediated in part by CREB transcription. We propose a model in which DP fusion hyperactivates the CREB signaling, possibly via CRTC activation [48] or suppression of the LKB1 signaling pathway [47], and ultimately leads to the aberrant elevation of LINC00473. This model merits further evaluation in future investigations of LINC00473 in FLC.

A major strength of our study is the combination of genome-scale and functional approaches, in patient samples and patient-derived disease models, to interrogate the significance of LINC00473 in FLC. Our analyses on matched patient samples reveal that the

expression levels of LINC00473 and the DP fusion are highly correlated in primary tumors and that this coordinate expression is likely restricted to tumor epithelial cells. The latter finding is particularly relevant given that the cell-based disease model in which we evaluate LINC00473 function is comprised primarily of FLC tumor epithelial cells from a patient-derived xenograft.

We highlight the role of LINC00473 in altering glycolysis and mitochondrial fitness in FLC, which is notable given the dearth of knowledge on both LINC00473-driven phenotypes and FLC metabolism. In our lab, we have observed low levels of spare respiratory capacity (SRC) in FLC cells (personal communication, Donald Long, Jr.), suggesting that FLC mitochondria cannot readily adapt to acute energy stressors [66]. In this study, we revealed that LINC00473 is at least partly responsible for the low SRC phenotype in FLC. We also uncovered that LINC00473 increases the expression of genes in metabolic pathways that are activated in FLC, including glutamine metabolism and proline synthesis. PYCR1 confers the final step of proline synthesis and is considered a pro-tumorigenic gene commonly overexpressed in cancers [67]. Studies have demonstrated that increased levels of mitochondrial redox facilitate PYCR1 to drive glutamine flux into proline synthesis [67,68]. Moreover, oxidative stress significantly reduces SRC [69]. Strikingly, our RNA-seq data reveals that PYCR1 is among the most prominently upregulated genes in FLC tumors as well as in LINC00473-overexpressing FLC cells, and is also significantly elevated in LINC00473-induced mouse neurons [55]. It may be the case that LINC00473 sustains high levels of basal respiration and subsequently increases reactive oxygen species abundance, compromising overall mitochondrial fitness.

A well-established challenge in the FLC field is the scarcity of disease models [4]. For a few years, the only available cell-based model of FLC was the DP fusion-expressing AML12 mouse hepatocyte line [27]. A major limitation of this model for evaluating the primate-specific LINC00473 is its murine origin. In more recent years, at least two other human cancer lines were engineered to express the fusion oncogene, both of which we confirm to exhibit elevated levels of LINC00473 [52,70]. While it has been shown that the exogenous introduction of LINC00473 in mouse neurons can modulate synaptic activity, it is unknown whether the transcriptional programs and signaling pathways critical for LINC00473-mediated cancer phenotypes are conserved in non-primate contexts [55,56].

The development of a DNAJB1-PRKACA inhibitor has encountered major challenges due to the inhibition of wild-type PKA, which serves critical roles in maintaining normal physiology in multiple tissues [17–19,71]. Here, we observed that DP fusion-positive FLC cells depend on LINC00473 for cell growth and survival. Given that LINC00473 expression is restricted to a few tissue types, we propose that LINC00473 is a candidate target for blocking FLC tumor growth as part of systemic combination therapy. RNAi-based therapies are currently in clinical trials for cancer and other diseases, and additional research may indicate their potential application for lncRNA inhibition [72,73].

A major next step in examining LINC00473 pertains to its molecular functions. For example, it would be interesting to interrogate whether LINC00473 localizes to specific areas of chromatin to regulate transcription in a targeted fashion, and what binding partners LINC00473 might have in other cell compartments. Further, it is important to uncover the downstream targets and effectors of LINC00473 in FLC and other cancers. Another fascinating area that warrants further investigation is the mechanism underlying LINC00473-regulated mitochondrial activity and cellular energetics in FLC. This is particularly interesting due to the growing body of evidence suggesting that altered mitochondrial dynamics are critical to FLC pathology, including aberrant mitochondrial fission [52,74,75].

Together, this study establishes LINC00473 as one of the most well-characterized FLC marker genes to date. We demonstrate that LINC00473 is an important downstream mediator

of the DP fusion oncoprotein. Its expression is consistently upregulated in primary FLC tumors and multiple disease models, and it is specifically expressed in tumor epithelial cells. Our gain- and loss-of-function approaches identify LINC00473 as a critical regulator of FLC growth and reveal its role in dysregulating energetics in FLC mitochondria. Importantly, our findings strongly support the potential utility of LINC00473 as a biomarker for DP fusion activity for cell-based drug screening efforts and as a candidate therapeutic target for FLC.

## Materials and methods

### Ethics statement

This study was conducted according to Institutional Review Board protocols 1811008421 (Cornell University), 33970/1 (Fibrolamellar Cancer Foundation) or 1765 (Fred Hutchinson Cancer Center). Informed written consent was obtained from all human subjects or from the subject's parent or legal guardian if the subject is a minor.

### 1. Primary tumor samples and PDX tumors

**FLC tumors and NML.** *Cohort 1*. Informed written consent was obtained from all human subjects or the subject's parent or legal guardian if the subject is a minor. FLC and non-malignant liver samples were collected from FLC patients according to Institutional Review Board protocols 1811008421 (Cornell University) or 33970/1 (Fibrolamellar Cancer Foundation) and provided by the Fibrolamellar Cancer Foundation. All samples were de-identified prior to shipment to Cornell University and were collected from both male and female subjects, and importantly, some samples were collected from the same patient.

*Cohort 2*. Informed written consent was obtained from all human subjects or the subject's parent or legal guardian if the subject is a minor. FLC and non-malignant liver samples were collected from female and male FLC patients according to Institutional Review Board protocol 1765.

### 2. PDX tumors

*Mice*. Animal care, facilities, procedures, and technical services used in this study comply with NIH regulations, the Cornell University Institutional Animal Care and Use Committee (protocol 2017–0035), and the Association for the Assessment and Accreditation of Laboratory Animal Care (AAALAC) International. All animal research space and supporting equipment are controlled by technicians, residents, and board-certified laboratory animal veterinarians of the Center for Animal Resources and Education (CARE).

*Mouse strains and husbandry*. NOD.Cg-*Prkdc$^{scid}$ Il2rg$^{tm1Wjl}$*/SzJ (NSG) mice catalog number 005557 were sourced from The Jackson Laboratory and bred at Cornell with the supervision of the Center for Animal Resources and Education (CARE) breeding program.

*PDX tumor maintenance and passaging*. At the time of passaging, tumors (diameter of 0.5cm) are transplanted subcutaneously into the flank of NSG mice (male and female, 6–10 weeks of age). Each mouse harbors a single tumor. Animals are housed in ventilated cages with *ad libitum* access to food and water. Mice are monitored for signs of distress, such as poor grooming and weight loss until the endpoint, which is established when PDX tumors reach a diameter of 3cm at any greatest dimension. Mice are subsequently euthanized for tumor collection, which is divided for propagating tumor passaging. Procedures are performed according to protocols approved by the Cornell University Institutional Animal Care and Use Committee.

## 3. RNA-seq, snATAC-seq and ChRO-seq

*RNA isolation*: For RNA purification from tissues, 15-30mg of flash-frozen patient samples and patient-derived xenograft tumors were pulverized using dry-ice chilled Bessman Tissue Pulverizer Thomas Scientific, 1210V37). Each sample was hammered four times with a one-quarter hammer turn until tissues became a fine powder and transferred to 1.5mL Eppendorf tubes containing 600uL lysis buffer (Total RNA Purification Kit, Norgen Biotek, 17200). They were incubated on ice for 30 minutes before homogenization using Polytron System PT 1200E for 30s. Samples were centrifuged at 14,000 RPM for 2 minutes, and the supernatant was collected as tissue homogenate in a clean 1.5mL tube for immediate RNA purification. The tissue pulverizer was cleaned with 70% ethanol and RNase Away and chilled in dry ice for 2 minutes between sample preparations. The polytron was cleaned in three washes of water and RNase Away between sample processing. According to the manufacturer's instructions, RNA was extracted from tissue homogenates and cells using the Total RNA Purification Kit (Norgen Biotek, 17200). RNA concentrations were measured using Nanodrop One, and RNA integrity was quantified using the 4200 Tapestation.

*RNA-seq*. Gene expression was analyzed in 35 FLC and 10 non-malignant liver (NML) samples. Of these, 16 FLC and 2 NML samples were published previously (accession: EGAS00001004169) [53]. The remaining 19 FLC and 8 NML samples represent a cohort of samples with new RNA-seq data provided by this study (accession: GSE233148) and were previously published with small RNA-seq data (accession: GSE181922). Consistent with previously published samples, RNA was isolated from the new cohort using the Total RNA Purification Kit (Norgen Biotek). Libraries were prepared using the NEBNext Ultra II Directional Library Prep Kit following PolyA enrichment at the Cornell Transcriptional Regulation and Expression Facility (Cornell University, Ithaca, NY). Libraries were then sequenced using the NextSeq500 platform (Illumina). To handle batch effects in primary patient tissues, we employed the limma package, a widely recognized tool for the analysis of high-dimensional genomic data as part of our analysis. Specifically, we utilized the limma::removebatcheffect function that fits a linear model to the data to batches of FLC or NML groups and subsequently removes batch effects from expression data.

Sequencing results were aligned to the human genome (hg38) using STAR (v2.4.2a) and quantified using Salmon (v0.8). Differential expression of genes was determined using DESeq2 (v1.29). Genes with an average expression of less than 5 (base mean) were discarded from the analysis. Enrichr was used for pathway analyses as described in Chen et al [62]. Our RNA-seq datasets on patient tissues, as well as LeGO-473ox and LeGO-Ctl cells, were deposited into the Gene Expression Omnibus (GEO: GSE233148). RNA-seq datasets on LeGO-473ox and LeGO0Ctl tumors can be found through GEO: GSE253465.

*snATAC-seq*: Single-nucleus ATAC (snATAC) analysis was performed as previously described [60]. Data generated from our snATAC-seq studies were previously published and are available at the Gene Expression Omnibus through accession number GSE202315.

*ChRO-seq*: Chromatin Run-On sequencing (ChRO-seq) analysis was performed as previously described [53]. Data can be downloaded from the European Genome-Phenome Archive through the European Genome-Phenome Archive via accession number EGAS00001004169.

## 4. RT-qPCR

RNA was extracted from tissue homogenates and cells using the Total RNA Purification Kit (Norgen Biotek, 17200) according to the manufacturer's instructions. RNA concentrations were measured using Nanodrop One and 0.25ug of RNA for gene analysis was used for reverse transcription using the High-Capacity RNA-to-cDNA Kit (Applied Biosystems, 4368814)

according to manufacturer instructions. Gene expression was quantified using the TaqMan Gene Expression Master Mix (ThermoFisher Scientific, Waltham, MA). Gene expression was normalized to RPS9 (assay ID: Hs02339424_g1). qPCR was performed using the BioRad CFX96 Touch Real-Time PCR Detection System (Bio-Rad Laboratories, Richmond, CA).

Gene Taqman assays:

*DNAJB1-PRKACA* (assay ID: AIHSPSH)

*PRKACA* (custom assay ID: AIS093C)

*DNAJB1* (assay ID: Hs00428680_m1)

*LINC00473 TV1* (custom assay ID: APT2A4G)

*LINC00473 TV2* (custom assay ID: APRWGJJ)

*U6* (assay ID: 001,973)

*tRNA-Glycine* (custom assay ID: CS1RUHV)

*CTNNB1* (assay ID: Hs00355045_m1)

*SLC16A14* (assay ID: Hs005431300_m1)

*VCAN* (assay ID: Hs0017642_m1)

*RPS9* (assay ID: Hs02339424_g1)

## 5. Subcellular Fractionation

FLC cells were washed and pelleted in PBS at $1 \times 10^6$ for subcellular fractionation as previously described [76]. Briefly, cell pellets were lysed in RLN1 buffer (50 mM Tris-HCl pH 8.0, 140 mM NaCl, 1.5 mM MgCl2, 0.5% NP-40, 40 units/mL RNAse inhibitor) on ice for 5 min and subsequently centrifuged at 500g for 2 min. The cytoplasm contained in the supernatant was isolated from the pelleted nuclear fraction. RNA was isolated from both fractions using Total RNA Purification Kit (Norgen Biotek, 17200) as per manufacturer instructions. RNA concentration was quantified using Nanodrop One. Equal amounts of RNA and cDNA were used for subsequent reverse transcriptase reactions and qPCR, as described above. All samples were tested for U6 (nuclear control) and tRNA-Glycine (cytoplasmic control). The sum of nuclear and cytoplasmic RNA expression levels of each gene was set to 100%, and the percentage of each transcript localized to each compartment was calculated. For U6 and tRNA-Glycine, reverse transcription was performed using 0.2ug of RNA and TaqMan MicroRNA Reverse Transcription Kit (ThermoFisher, 4311235), followed by qPCR using TaqMan Universal PCR Master Mix (ThermoFisher, 4324018). Data represents 3 biological replicates.

## 6. Cells

*FLC cells.* FLC cells were previously described by Francisco et al., generated from a patient-derived xenograft model developed by Oikawa et al. and were a generous gift from Dr. Nabeel Bardeesy [61,77]. FLC cells were grown in complete RPMI media (RPMI 1640, Gibco 11875119; 10% FBS, heat-treated at 56˚C for 30 minutes, Gibco 26140079; 1% penicillin-streptomycin, Gibco 15140122; and 25ng/mL human hepatic growth factor, Gibco PHG0321).

*LINC00473 Overexpression and Knockdown cell lines*: To enable LINC00473 knockdown via lentiviral integration of shRNA, our collaborators in the Wu Lab at the University of Florida generously gifted us lentiviral vector harboring short hairpin RNAs (shRNAs) targeting

LINC00473 (sh473-2, sh473-4) and control (shCtl) that were previously used to downregulate gene expression in human lung cancer cell lines [47]. The pLKO.1 cloning vector served as the plasmid backbone, which is a replication-incompetent lentiviral vector encoding a puromycin resistance marker [78]. Two independent shRNA oligos targeting LINC00473 were individually cloned into pLKO.1 at the EcoRI and AgeI restriction sites, and subsequently transformed and replicated in DH5-Alpha E. coli cells under ampicillin selection. Next, the oligo insert was screened following restriction digest with EcoRI and NcoI for gel electrophoresis and sanger sequencing using the pLKO.1 sequencing primer. Next, lentiviral particles were produced by transfecting the pLKO.1 shRNA plasmid combined with the psPAX2 packaging plasmid and the pMD2.G envelope plasmid into HEK293T cells. Over 4 days, virus-containing media was harvested for virus isolation using Lenti-X concentrator, Takara) instead of ultracentrifugation. After performing puromycin kill curve assays in FLC and HEK293-DP cells, lentiviral particles were titered in a range of dilutions in DMEM complete (DMEM with 10% FBS and 2% Glutamax) and 10ug/mL polybrene and subsequently introduced to target cells using reverse transduction in order to generate a Poisson distribution curve of a viral multiplicity of infection (MOI). An MOI of 0.3 was selected for viral transduction to achieve a predicted single viral integration in 22% of treated cells while reducing the cell population with >1 integration to only 4%. Following puromycin selection, the majority of transduced FLC cells will express a single copy of shRNA to mitigate off-target effects, enabling the selection of a stable polyclonal cell culture. Polyclonal cells were grown across 5 passages and verified for stable gene knockdown using RT-qPCR. Conditioned medium was collected during cell growth. Next, monoclonal cell populations were generated by limiting dilution by seeding an average of 0.5 cells per well in 96-well plates in the presence of conditioned medium and puromycin. This ensures that some wells receive a single cell while minimizing the chance that any well is seeded with >1 cell. Cells were grown undisturbed for 14 days, and wells with expanded cell populations were transferred to larger culture dishes. Each monoclone was subsequently expanded, and gene knockdown was validated via RT-qPCR. Monoclonal cell lines with robust LINC00473 downregulation were selected for functional in vitro and in vivo assessments.

To develop stable LINC00473 overexpression via lentiviral integration of cDNA, the LeGO-lnc cloning vector served as the plasmid backbone (a gift from Dr. Jan-Henning Klusmann, RRID: Addgene plasmid_80624), which is a replication-incompetent lentiviral vector encoding a blasticidin resistance marker. A double-stranded cDNA clone of LINC00473 was designed based on the human sequence from the UCSC genome browser and subsequently synthesized and purchased from Integrated DNA Technologies (gBlocks Gene Fragment). The cDNA was cloned into LeGO-lnc at the NotI and SFFV restriction sites and subsequently transformed and replicated in DH5-Alpha E. coli cells under ampicillin selection. Next, the cDNA oligo insert was screened following restriction digest with NotI and PacI for gel electrophoresis and Sanger sequencing using the SFFV forward primer and MSCV reverse primer. Next, lentiviral particles were produced by transfecting the LeGO-LINC00473 plasmid combined with the psPAX2 packaging plasmid and the pMD2.G envelope plasmid into HEK293T cells. Over 4 days, virus-containing media was harvested for virus isolation using Lenti-X concentrator (Takara) instead of ultracentrifugation. The methods performed for generating polyclonal and monoclonal cell lines harboring LINC00473 overexpression and control are described above; however, antibiotic selection was enabled using blasticidin.

*HepG2-DP and HepG2-Ctl*: HepG2-DP cells were previously described using the HepG2 cell line (RRID: CVCL_0027) and were a generous gift from Dr. Sean Ronnekleiv Kelly [70]. Briefly, the *DNAJB1-PRKACA* expressing HepG2 cells (HepG2-DP) were generated using a mammalian dual-guide RNA-CRISPR-CAS9-EGFP vector (VectorBuilder, Chicago, IL) with guide RNA sequences targeting the first intronic regions of *DNAJB1* and *PRKACA* (gRNA1

(DNAJB1) 5'-CAGGAGCCGACCCCGTTCGT-3', gRNA2 (PRKACA): 5'-GTAGACGCG GTTGCGCTAAG-3'). Genomic DNA and mRNA were used to confirm the presence of *DNAJB1-PRKACA* fusion, as previously described [70]. Cells were grown in DMEM complete media (DMEM, high glucose, Gibco, 11965092; 10% FBS, heat-treated at 56C for 30 minutes, Gibco, 26140079; 5% penicillin-streptomycin 10,000U/mL, Gibco, 15140122; 5% Glutamax 100X, Gibco, 35050061).

*HEK293-DP and HEK293-WT*: HEK293-DP cells expressing DNAJB1-PRKACA have been previously described using the HEK293T cell line (RRID: CVCL_00663) and were a generous gift from Dr. Khashayar Vikili [52]. Cells were grown in DMEM complete media (DMEM, high glucose, Gibco, 11965092; 10% FBS, heat-treated at 56C for 30 minutes, Gibco, 26140079; 5% penicillin-streptomycin 10,000U/mL, Gibco, 15140122; 5% Glutamax 100X, Gibco, 35050061; 5% sodium pyruvate 100mM, Gibco, 11360070).

## 7. In vitro assays on cell growth, colony growth, viability, and apoptosis

*Cell growth curve*. Lentiviral-transduced FLC cells were seeded in a 24-well plate at a density of 50,000 cells per well (Day 0). Daily counts on total cell number were performed for 6 days in approximate 24-hour intervals after initial seeding. Cells were washed in PBS (calcium and magnesium-free; Gibco, 10010023) and coated with 200uL 0.25% trypsin-EDTA (Gibco, 25200056) followed by 5 min incubation at 37°C or until 90% of cells detached from the well. Various volumes of pre-warmed complete RPMI media (RPMI 1640, Gibco 11875119; 10% FBS, heat-treated at 56°C for 30 minutes, Gibco, 26140079; 1% penicillin-streptomycin, Gibco 15140122) were used to resuspend cells, specifically, 200uL on Day 1; 300uL on Day 2; 800uL on Day 3, 1600uL on Day 4, and 2000uL on Days 5–6. 10uL of cell suspension was transferred to a hemocytometer (Bright-Line, Z359629) and observed under a microscope under a 10X objective lens. Cells were counted from the large, central gridded square ($1mm^2$) and multiplied by $10^4$ to calculate the number of cells per mL, and subsequently multiplied by the total volume of the cell resuspension (mixture of 0.5% trypsin-EDTA and complete media; ranges from 400uL to 2200uL) to determine the total number of cells per well. Cells were prepared for counting in groups of 4 wells to avoid over-trypsinization and clumping. The protocol for evaluating cell growth in transduced HEK293-DP cells was performed as previously reported by Kim et al. [52]. Doubling time was determined using the following formula, such that initial and final cell concentrations were from days 1 and 5, respectively. Duration was 96 hours.

$$\text{Doubling time} = \frac{\text{Duration} \times ln(2)}{ln\left(\frac{\text{Final concentration}}{\text{Initial concentration}}\right)}$$

*Soft agar assay*. 6-well plates were coated with 0.6% UltraPure Low Melting Point Agarose (Invitrogen, 16520050) mixed with complete RPMI media (RPMI 1640, Gibco 11875119; 10% FBS, heat-treated at 56C for 30 minutes, Gibco, 26140079; 1% penicillin-streptomycin, Gibco 15140122). Transduced FLC cells were suspended in 0.3% agarose and seeded in 6-well plates at a density of 10,000 cells per well. Once solidified, an additional layer of 0.3% agarose was added to coat each well. Plates were incubated at 37C for a colony growth period of 8 days. Cells were fixed and stained with 200uL of 1 mg/mL solution of Nitro Blue Tetrazolium Chloride (Invitrogen, N6495). Six biological replicates were performed, and 10–12 independent fields of view were imaged per well. Colony area was quantified using FIJI software. Data represents the cell colony growth as a percent of the total area per image.

*EdU-incorporation assay*. 72 hours after seeding in 96-well plates at a density of 3500 cells per well, cells were incubated with 10 μM EdU at 37°C in complete media for 2 h. Cells were then fixed with 4% paraformaldehyde for 20 min at room temperature and permeabilized

using 0.5% Triton X-100 in PBS for 20 min. The Invitrogen Click-iT Plus EdU AlexaFluor 488 Imaging Kit (Invitrogen, Waltham, MA, C10637) was used to detect EdU according to the manufacturer's instructions. Nuclei were stained using DAPI (ThermoFisher, D1306) and imaged using ZOE Fluorescent Cell Image (Bio-Rad Laboratories, Richmond, CA). Images were analyzed using FIJI. For EdU-positive cells, the threshold value was set to 10. For analyzing particles, counted those particles with size = 250-Infinity and circularity = 0.4–1.

*TUNEL assay.* 72 hours after seeding in a 96-well plate at a density of 3500 cells per well, cells were washed twice with PBS and fixed using 4% paraformaldehyde for 15 min at room temperature. Permeabilization was performed by using 0.5% Triton X-100 in PBS for 20 min. Cells were washed twice with deionized water. Positive control wells were treated with 1X DNase I, Amplification Grade (ThermoFisher, 18,068–015) solution according to the manufacturer's instructions. Labeling and detection of apoptotic cells were completed using the Invitrogen Click-iT Plus TUNEL Assay for In Situ Apoptosis Detection 488 kit (Invitrogen, Waltham, MA, catalog #: C10617) according to the manufacturer's instructions. Nuclei were stained using DAPI (ThermoFisher, D1306) and imaged using ZOE Fluorescent Cell Image (Bio-Rad Laboratories, Richmond, CA). Images were analyzed using FIJI. For TUNEL-positive cells, the threshold value was set to 14. For analyzing particles, counted those particles with size = 250-Infinity and circularity = 0.4–1.

## 8. Subcutaneous cell transplantation and tumor growth

FLC monoclonal cell lines overexpressing LINC00473 (LeGO-473ox) or empty vector control (LeGO-Ctl) were established, as described above. Expression of key genes of interest, including *DNAJB1-PRKACA*, *LINC00473*, and *RPS9* control, was assessed via RT-qPCR (as previously described) across 5 independent passages per monoclone to validate stable gene expression. Two weeks prior to transplantation, cells were grown in absence of antibiotics and submitted for mycoplasma testing. Cells were washed and resuspended in ice-cold PBS and mixed with an equal volume of matrigel for single-flank injections of 2 million cells in 200uL per animal. Mice were monitored twice a week to monitor for palpable tumors, and after initial tumor detection (denoted as Day 0), tumor volumes were quantified three times a week using digital calipers. Mice were euthanized, and tumors were collected 7 days after initial tumor detection. Relative tumor % growth is calculated as (tumor volume at Day N/the tumor volume at Day 0) x 100%.

## 9. Western Blot

FLC cells were lysed in RIPA buffer containing Halt Protease and Phosphatase Inhibitor Cocktail (ThermoFisher, 78441) at 4°C. Cells were incubated for 30 minutes and centrifuged at 14,000g for 10 minutes at 4°C. Total protein in the supernatant was quantified using the Pierce BCA Protein Assay Kit (ThermoFisher, 23225). Samples were denatured in NuPAGE LDS Sample Buffer (ThermoFisher NP0007) containing 5% β-Mercaptoethanol for 10 minutes at 70°C and loaded to a 10% NuPAGE gel (ThermoFisher, NP0301BOX). After electrophoresis, samples were transferred to polyvinylidene difluoride membranes and blocked in 3% bovine serum albumin (BSA) in tris-buffered saline and 0.5% Tween 20 (TBST) for 1 hour at room temperature. Membranes were incubated in primary antibodies at 4C overnight. Membranes were subsequently incubated in secondary antibodies at room temperature for 1 hour followed by three washes of TBST. Immunoblots were visualized using an enhanced chemiluminescence (ECL) kit (Cytiva, 89238–012) and a ChemiDoc MP (Bio-Rad).

The antibodies used in this study consisted of PRKACA (1:1000 dilution, Cell Signaling Technology, 4782, RRID:AB_2170170), HSP40 (1:1000 dilution, Santa Cruz Biotechnology, sc-398766, RRID:AB_2941764), Vinculin (1:1000 dilution, Santa Cruz Biotechnology, sc-

73614, RRID:AB_ 2941767), mouse IgG secondary antibody (1:2000, Santa Cruz Biotechnology, sc-525409, RRID:AB_2941766), and rabbit IgG secondary antibody (1:2000, Cell Signaling Technology, 7074, RRID:AB_2099233). Protein quantification was performed using FIJI software (normalized to vinculin; loading controls were run on the same blot).

## 10. siRNAs

*siRNA synthesis and LNP formulation.* All oligonucleotides were prepared using commercially available 5′-O-DMT-3′-O-(2-cyanoethyl-N,N-diisopropyl) phosphoramidite monomers by following standard protocols for solid phase synthesis and deprotection [79,80]. The PS linkages were introduced by phosphite oxidation utilizing 0.1 M N,N-dimethyl-N'-(3-thioxo-3H-1,2,4-dithiazol-5-yl)methanimidamide (DDTT) in pyridine. The *N*-acetylgalactosamine (GalNAc) ligand was introduced to the 3′-end of the sense strand of the siRNA using a functionalized solid support as described [81]. 2'-O-hexadecyl (C16) and cholesterol (chol)-functionalities were introduced as described previously [82,83]. After deprotection, single strands were purified by ion-exchange HPLC, followed by desalting and annealing of equimolar amounts of complementary strands to provide the desired siRNAs and siRNA conjugates. Lipid nanoparticles (LNPs) containing encapsulated siRNAs were formulated as described previously [84].

*siRNA in vitro activity by transfection.* siRNA efficacy for gene silencing was initially evaluated by reverse transfection on FLC cells in 48-well plates. 20 uL solutions of siRNA were prepared at 10X concentration in PBS and subsequently mixed with OptiMEM (Gibco 31985062) and Lipofectamine RNAiMax Transfection Reagent (Invitrogen 13778030) according to manufacturer instructions. After 30 minutes of incubation at room temperature, the mixture was transferred to wells. Next, FLC cells were plated at a density of 34000 cells per well for a final volume of 200 uL. The final concentration of siRNAs was 50 nM. Plates were gently rocked by hand from side to side after plating and incubated at 37C for 48 hours before cell preparation for RNA isolation.

Transfection of siPKA and non-targeting controls in FLC cells was performed using siGENOME single and SMARTpool siRNA PRKACA from Dharmacon. For siRNA treatment, 12 μl of 20 μM siRNA was added to the cells with Lipofectamine RNAiMAX reagent in Opti-MEM, incubated for 72 hr, and harvested for subsequent RNA isolation.

*Free-Uptake.* To test the efficacy of different siRNA delivery strategies in FLC cells, we performed free uptake studies in FLC cells with CTNNB1-targeting siRNAs that were conjugated to N-acetylgalactosamine (GalNAc), 2′-O-hexadecyl (C16), cholesterol (chol), or encapsulated LNPs. 20 uL solutions of siRNA were prepared at 10X concentration and transferred to wells. FLC cells were plated at a density of 34000 cells per well for a final volume of 200 uL. Plates were gently rocked by hand from side to side after plating and incubated at 37C for 48 hours. The final concentrations of GalNAc-, C16-, and chol-conjugated siRNAs were 50nM and 500nM. The final concentrations of LNP-siRNAs were 2.5nM, 12.5nM, 62.5nM, and 312.5nM.

To test gene silencing of the DP fusion using siDP#1-LNP and siDP#2-LNP, free-uptake was performed on FLC cells as described above using 6-well plates for a final volume of 2mL and final siRNA concentration of 5nM at 37C for 96 hours. At the end of incubation, a portion of the cell suspension was allocated for RNA purification, and the remainder was used for protein isolation. Data shown represent at least 3 biological replicates.

## 11. Seahorse assay

Extracellular acidification rate (ECAR) measures pH changes as cells produce lactate and thus serves as a proxy for glycolysis. Cells were plated on XFe24 cell culture plates at the following

seeding densities: A3–91,000 cells per well; C2–60,000 cells per well; D3–57,000 cells per well; F6–81,000 cells per well, and incubated for 48-hour at 37˚C with 5% $CO_2$. Following incubation, cells were washed in PBS, and culture media was replaced with unbuffered Dulbecco's Modified Eagle's Medium (DMEM) supplemented with 2 mM L-glutamine. The compound concentrations for the mitochondrial stress test were as follows (final concentration): Port A: Glucose (11 mM), Port B: Oligomycin (1 μM), and Port C: 2-Deoxy-D glucose (50 mM). The ECAR was normalized to the total cell number using the Celigo image cytometer. Respirometry data were collected using Agilent Wave v2.4 software and were expressed as mean ± SEM. To evaluate glucose metabolism, ECAR is measured in response to the addition of D-glucose to activate glycolysis, oligomycin to shut down oxidative phosphorylation and subsequently stimulate maximal glycolytic capacity, and deoxy-D-glucose (2-DG), a competitive inhibitor of glucose that blocks glycolysis. Glycolytic reserve capacity was calculated by subtracting glycolysis from maximum glycolytic capacity.

Oxygen consumption rate (OCR) was measured by the Agilent Seahorse XFe 24 Bioanalyzer. Cells were plated on XFe24 cell culture plates at the following seeding densities: A3–91,000 cells per well; C2–60,000 cells per well; D3–57,000 cells per well; F6–81,000 cells per well, and incubated for 48-hour at 37˚C with 5% $CO_2$. Following incubation, cells were washed in PBS, and culture media was replaced with unbuffered Dulbecco's Modified Eagle's Medium (DMEM) supplemented with 4.5 g/L glucose, 4 mM glutamine, and 1 mM pyruvate. The compound concentrations for the mitochondrial stress test were as follows (final concentration): Port A: Oligomycin (1 μM), Port B: Carbonyl cyanide-4 (trifluoromethoxy) phenylhydrazone (FCCP) (1 μM), Port C: Rotenone/Antimycin A (2 μM) each. The OCR was normalized to the total cell number using the Celigo image cytometer. Respirometry data were collected using Agilent Wave v2.4 software and were expressed as mean ± SEM. To evaluate mitochondrial respiration, OCR is measured in response to a series of modulators that alter electron transport chain function. Specifically, treatment with oligomycin inhibits ATP synthase and subsequently blocks respiration. Next, carbonyl cyanide 4-trifluoromethoxyphenylhydrazone (FCCP) acts as an uncoupling agent that stimulates the ETC to operate at maximum capacity, revealing maximum respiratory capacity. Lastly, antimycin A, rotenone (R/A), complex III and I inhibitors are added to shut down ETC function, uncovering non-mitochondrial respiration. Spare Respiratory Capacity (SRP) is calculated by subtracting basal respiration from maximum respiratory capacity.

## Supporting information

**S1 Fig. LINC00473 is a distinct transcription unit in FLC tumors. (A)** Genome snapshot of the LINC00473 and *PDE10A* loci on chr6:165250000–166050000. Transcriptional signal on the plus and minus strand are shown in red and blue, respectively, in FLC tumors and non-malignant liver (NML) tissues. The RefSeq annotations for LINC00473 TV1 and TV2, and *PDE10A*, are shown on the bottom panel. Both genes are transcribed on the minus strand. LINC00473 has markedly higher levels of gene body transcription than the space in between LINC00473 and *PDE10A*. **(B)** Lists of primary patient tumors used in RNA-seq counts in a patient-matched subset from Cohort 1, gene expression via RT-qPCR in a patient- matched subset from Cohort 1, and correlation analyses on RNA expression levels of the DP fusion and LINC00473 isoforms TV1 and TV2 using patient tumors from Cohort 1, as described in Fig 1D, 1E, and 1F, respectively. Tumors indicated in red are common for all three analyses, and tumors highlighted in blue are shared for Fig 1E and 1F. **(C)** Normalized levels of DP fusion and LINC00473 isoforms TV1 and TV2 RNA in HEK293-DP cells relative to wild-type control (HEK293-Ctl). Normalized levels are 2^(-dCt) values using RPS9 for normalization

and presented from 3 technical replicates. Data are represented as mean ± SD. P values are calculated by 2-tailed Student's t-test. *p < 0.05, **p < 0.01, ***p < 0.001. **(D, E)** Pathway enrichment analyses of significantly upregulated genes (n = 1666) (A) and downregulated genes (n = 1497) (B), respectively, in primary FLC tumors (n = 35) relative to non-malignant tissue (n = 10). Genes filtered for expression with base mean > 100, log2FC > 1 or < -1, and padj < 0.05 (DESeq). Pathways with adjusted p-value < 0.05 represented in figure. Color intensity represents odds ratio value.
(PDF)

**S2 Fig. LINC00473 is enriched in FLC tumor epithelial cells. (A)** snATAC-seq UMAP plot demonstrating eight cell clusters found in primary FLC, metastatic FLC, and NML tissue (~9500 nuclei). **(B)** Single-nucleus analysis of chromatin accessibility near the LINC00473 locus. Increasing signal is indicated by the color gradient (maximum signal is dark purple and minimal signal is light blue). **(C)** Top panel depicts the gene body of LINC00473, which is transcribed on the minus strand. Bottom panel demonstrates the quantification of snATAC-seq chromatin accessibility signal near the LINC00473 locus in different cell types. **(D)** Boxplot showing normalized counts of chromatin accessibility signal of LINC00473 in distinct cell types. **(E)** Subcellular fractionation followed by RT-qPCR in FLC cells using TaqMan primers designed to detect variants 1 (TV1) and 2 (TV2) of LINC00473, U6 (nuclear control) and tRNA-Gly (cytoplasmic control). Data represents n = 3 biological replicates ± SD. For A-D, data are represented from n = 1 FLC tumor, 1 Metastatic tumor and 1 NML.
(PDF)

**S3 Fig. Silencing of the DP fusion. (A, C)** Representative immunoblot of protein expression of DNAJB1- PRKACA (DP) fusion is detected with a protein kinase A catalytic α subunit (PKA) antibody. WT PKAc, DP fusion major, and DP fusion minor are identified. Lane 1, siDP#1-LNP; Lane 2, siDP#2-LNP; Lane 3, siLuciferase (siLuc-LNP) negative control; Lane 4, mock negative controls following 1.25nM treatment (A) or 2.50nM treatment (C) with siRNA-LNPs or mock condition overs 96 hours. Vinculin loading control is shown in the lower panel and run on the same blot. **(B, D)** Fold change of protein levels of the blot in panel A (B) and panel C (D), relative to siLuc negative control (n = 3). **(E)** Representative immunoblot of protein expression of WT DNAJB1. Lane 1, siDP#1-LNP; Lane 2, siDP#2-LNP; Lane 3, siLuciferase (siLuc-LNP) negative control; Lane 4, mock negative control. siRNA-LNP treatments at 5nM, or mock condition, over 96 hours. Vinculin loading control is shown in the lower panel and run on the same blot (n = 3). **(F)** Gene expression from RT-qPCR following free uptake of siDP#1-LNP, siDP#2-LNP, and siLuc-LNP at 5nM treatment over 96 hours in FLC cells, as shown in Fig 3F (n = 3). Data are represented as mean ± SD. P values are calculated by 2-tailed Student's t-test. *p < 0.05, **p < 0.01, ***p < 0.001.
(PDF)

**S4 Fig. LINC00473 is efficiently downregulated or overexpressed in FLC and HEK293-DP cells. (A)** LINC00473 TV1 expression from RT-qPCR in FLC monoclonal cells following limiting dilutions of polyclonal cell lines to isolate single-clone cell colonies with stable gene knockdown using two independent shRNAs (sh473-2, sh473-4) or non-targeting shRNA control (shCtl) (n = 3), or overexpression using cDNA plasmid encoding LINC00473 TV1 (LeGO-473ox) or empty- vector control (LeGO-Ctl) (n = 3). **(B)** Expression of LINC00473 TV2 was queried via RT-qPCR in FLC monoclones described in panel A (n = 3). **(C)** LINC00473 TV2 expression from RT-qPCR in FLC polyclones induced with LINC00473 TV1 overexpression (n = 3). **(D)** Expression of LINC00473 TV1 from RT-qPCR in HEK293-DP polyclonal cells following lentiviral transfection enabling stable gene knockdown using two

independent shRNAs (sh473-2, sh473-4) or non-targeting shRNA control (shCtl) (n = 3) or overexpression using cDNA plasmid encoding LINC00473 TV1 (LeGO-473ox) or empty-vector control (LeGO-Ctl) (n = 3). The expression of second variant of LINC00473 (TV2) in HEK293-DP cells with stable sh473-2, sh473-4, or shCtl (n = 3). **(E)** Cell growth curve of HEK293-DP monoclonal cells with stable LINC00473 knockdown (sh473-2: clones F3, F4; sh473-4: clones G1, G3) and non-targeting control (shCtl: clones H4, H5). Each monoclonal cell line was quantified 2 times across independent passages, and each point is the average cell count of 3 replicates. **(F)** Expression levels of both LINC00473 isoforms from RT-qPCR in HEK293-DP monoclones with stable gene knockdown using two independent shRNAs (sh473-2, sh473-4) or non-targeting shRNA control (shCtl). Data are represented as mean across 3 replicates ± SD. P values are calculated by 2-tailed Student's t-test. *p < 0.05, **p < 0.01, ***p < 0.001.
(PDF)

**S5 Fig. LINC00473 upregulates genes enriched in metabolism pathways and downregulates pathways related to apoptosis. (A)** Normalized counts of several genes of interest from RNA-seq of LeGO-473ox and LeGO-Ctl control FLC monoclonal cells. Bars are ± SD. The panel includes FLC-relevant transcriptional regulators (*MYC*), or genes associated with cellular metabolism (*IDH1*, *PYCR1*) and cell survival (*XIAP*, *CDK6*, *MALAT1*). **(B)** Pathway analyses using the 1403 upregulated genes in FLC monoclones with LINC00473 overexpression (LeGO-473ox) relative to empty vector control (LeGO-Ctl). Genes were filtered for expression with base mean > 100, log2FC > 1 and padj < 0.05 (DESeq). **(C)** Pathway analyses using the 1374 downregulated genes in FLC monoclones with LeGO-473ox relative to LeGO-Ctl. Genes were filtered for expression with base mean > 100, log2FC < 1 and padj < 0.05 (DESeq). **(D, E)** Gene list overlap analysis using significantly up- (D) or down-regulated (E) genes in FLC tumors relative to NML (n = 1497), and in LeGO-473ox cells relative to control (n = 1374). Pathways with p-value < 0.05 represented in figure. Color intensity represents odds ratio value. P values are calculated by 2-tailed Student's t-test. *p < 0.05, **p < 0.01, ***p < 0.001.
(PDF)

**S6 Fig. Gene expression in tumors generated from LINC00473 overexpressing cell implantation. (A)** Expression of FLC-related genes from RT-qPCR in LINC00473-overexpression (LeGO-473ox) and empty-vector control (LeGO-Ctl) subcutaneous FLC tumors. **(B)** KEGG pathway analyses using the 955 upregulated genes in FLC monoclones with LINC00473 over-expression (LeGO-473ox) relative to empty vector control (LeGO-Ctl). Genes were filtered for expression with base mean > 100, log2FC > 1 and padj < 0.05 (DESeq). **(C)** KEGG pathway analyses using the 1468 downregulated genes in FLC monoclones with LeGO-473ox relative to LeGO-Ctl. Genes were filtered for expression with base mean > 100, log2FC < 1 and padj < 0.05 (DESeq). Pathways with p-value < 0.05 represented in figure. Color intensity represents odds ratio value. P values are calculated by 2-tailed Student's t-test. *p < 0.05, **p < 0.01, ***p < 0.001.
(PDF)

**S1 Table. Primary tissue information.** Information on primary patient samples including tumor (FLC) and normal/non- malignant liver (NML) tissues. Institutions that performed bulk RNA-sequencing from Cornell University or Fred Hutchinson Cancer Center (FHCC) are listed.
(XLSX)

**S2 Table. Most significant differentially expressed genes in FLC versus non-malignant liver.** Genes significantly differentially expressed in primary FLC tumors (n = 35) versus non-

malignant liver (NML) (n = 10). Genes filtered for expression with base mean > 1000, log2FC > 1 or < -1, and padj < 0.05 (DESeq).
(XLSX)

**S3 Table. Most significant differentially expressed genes in LINC00473 overexpressing cells (LeGO-473ox) versus control cells (LeGO-Ctl).** Differentially expressed genes in FLC cells that stably overexpress LINC00473 (LeGO-473ox, n = 2) relative to empty vector control (LeGO-Ctl (n = 2). Genes filtered for expression with base mean > 100, log2FC > 0.5 or <-0.5, and padj < 0.05 (DESeq).
(XLSX)

**S4 Table. Most significant differentially expressed genes in LINC00473 overexpressing tumors (LeGO-473ox) versus control tumors (LeGO-Ctl).** Differentially expressed genes in LINC00473 overexpressing tumors (LeGO-473ox, n = 4) relative to empty vector control tumors (LeGO-Ctl, n = 4). Genes filtered for expression with base mean > 100, log2FC > 0.5 or <-0.5, and padj < 0.05 (DESeq).
(XLSX)

## Acknowledgments

We are grateful for the support provided by Rishi Puri and Scott Butler at the PATh PDX Facility and the Cornell Transcriptional Regulation and Expression Facility at Cornell University (Cornell University, Ithaca, NY). We thank past and current members of the Sethupathy lab for helpful discussions and support and Jonathan Villanueva for assistance in the computational analysis of ChRO-seq. We thank the Alnylam bioinformatics, high-throughput synthesis and RNAi lead discovery team for designing, synthesizing and screening the siRNAs. We thank Nabeel Bardeesy at the Cancer Center of Massachusetts General Hospital and Harvard Medical School for his generous gift of the FLC cell line. We are grateful to the FLC patient donors whose contributions make this work possible. Figure schematics were created with BioRender.com.

## Author Contributions

**Conceptualization:** Rosanna K. Ma, Praveen Sethupathy.

**Data curation:** Rosanna K. Ma, Pei-Yin Tsai.

**Formal analysis:** Rosanna K. Ma, Pei-Yin Tsai, Alaa R. Farghli, Alexandria Shumway, Matt Kanke, John D. Gordan, Taranjit S. Gujral, Manabu Nukaya, Leila Noetzli, Sean Ronnekleiv-Kelly, Wendy Broom, Joeva Barrow, Praveen Sethupathy.

**Funding acquisition:** Rosanna K. Ma, Khashayar Vakili, Praveen Sethupathy.

**Investigation:** Rosanna K. Ma, Praveen Sethupathy.

**Methodology:** Rosanna K. Ma, Praveen Sethupathy.

**Project administration:** Rosanna K. Ma, Praveen Sethupathy.

**Resources:** Rosanna K. Ma, John D. Gordan, Taranjit S. Gujral, Khashayar Vakili, Sean Ronnekleiv-Kelly, Wendy Broom, Praveen Sethupathy.

**Supervision:** Praveen Sethupathy.

**Validation:** Rosanna K. Ma, Pei-Yin Tsai.

**Visualization:** Rosanna K. Ma.

**Writing – original draft:** Rosanna K. Ma, Praveen Sethupathy.

**Writing – review & editing:** Rosanna K. Ma, Pei-Yin Tsai, Alaa R. Farghli, Alexandria Shumway, Matt Kanke, John D. Gordan, Taranjit S. Gujral, Khashayar Vakili, Manabu Nukaya, Leila Noetzli, Sean Ronnekleiv-Kelly, Wendy Broom, Joeva Barrow, Praveen Sethupathy.

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
