## [Decision Letter · Decision Letter 0]

1 Nov 2023

Dear Dr Sethupathy,

Thank you very much for submitting your Research Article entitled 'DNAJB1-PRKACA fusion protein-regulated LINC00473 promotes tumor growth and alters mitochondrial fitness in fibrolamellar carcinoma' to PLOS Genetics.

The manuscript was fully evaluated at the editorial level and by independent peer reviewers. The reviewers appreciated the attention to an important problem, but raised some concerns about the current manuscript. Based on the reviews, we will not be able to accept this version of the manuscript, but we would be willing to review a revised version. We cannot, of course, promise publication at that time.

If you decide to revise the manuscript for further consideration at PLOS Genetics, please aim to resubmit within the next 60 days, unless it will take extra time to address the concerns of the reviewers, in which case we would appreciate an expected resubmission date by email to plosgenetics@plos.org.

Please do not hesitate to contact us if you have any concerns or questions.

Yours sincerely,

John Prensner

Guest Editor

PLOS Genetics

Gregory Barsh

Editor-in-Chief

PLOS Genetics

The reviewers all find this manuscript potentially appropriate for PLOS Genetics. Please consider the reviewer comments and plan to address each comment in a point-by-point response if you choose to resubmit your work.

Reviewer's Responses to Questions

**Comments to the Authors:**

Reviewer #1: In their manuscript entitled, “DNAJB1-PRKACA fusion protein-regulated LINC00473 promotes tumor growth and alters mitochondrial fitness in fibrolamellar carcinoma”, Ma and colleagues investigate the mechanism of action of LINC00473 in the context of the DNAJB1-PRKACA fusion that drives fibrolamellar carcinoma (FLC), a rare but deadly cancer.

The authors generate primary RNA-Seq data from FLC patients and matched non-malignant liver samples. This dataset is a valuable community resource given the rare nature of this cancer type, and the authors have duly deposited the data into GEO and other repositories. Next, the authors establish using CRISPR-Cas9 deletion models as well as DNAJB1-PRKACA knockout models that LINC00473 is regulated by the fusion protein. They then perform gain-of-function and loss-of-function studies followed by RNA-Seq and molecular pathway analysis to predict affected biological pathways. Finally, an exploration of oncogenic mechanisms is explored using metabolic assays and xenograft models.

The manuscript is well written and clear. The experiments appeared to be well designed, and the data are generally convincing.

My biggest criticism of the manuscript in its current form is that I believe more could be done to build a case that LINC00473 is the major downstream target of the DNAJB1-PRKACA fusion. In the current manuscript, the authors cite literature evidence and prior work showing that LINC00473 has already been established as an oncogene. However, the authors generate a cohort of RNA-Seq from FLC and NML that could potentially be used in a hypothesis generating fashion. How do we know that LINC00473 is the most important target of the DNAJB1-PRKACA fusion protein? What other oncogenic mechanisms might be at play here?

These questions are not meant to detract from the well-designed experiments that the authors perform in the manuscript. However, a more thorough RNA-Seq analysis would enhance the manuscript’s overall impact. Some of these thoughts are echoed below as specific points.

Thank you to the reviewers and editors for the opportunity to serve as a reviewer.

Major points:

- It appears that the RNA-Seq dataset is being used in a hypothesis-driven rather than a hypothesis generating manner. The authors cite the literature support for LINC00473 to motivate the study as it has been studied in several cancer contexts, but the RNA-Seq analysis (Figure 1) indicates that LINC00473 is but one of the top 100 or so most highly overexpressed genes when comparing FLC versus NML. What role do the other upregulated/downregulated genes have in the biology of FLC? Are there other hypotheses? Are these differentially expressed genes related to the same molecular pathways? This appears to be partially addressed in Figure 4D, but it appears that there may be opportunities to better utilize the RNA-Seq cohort.

- One thought would be to create gene sets based on the differentially expressed genes between FLC and NML. Those gene sets can be further analyzed for molecular pathways, etc. In addition, PCA or other dimensionality reduction analysis may show biologically relevant patterns in the data. It appears that the RNA-Seq data came from multiple batches. Has any effort been taken to assess for batch effects?

- Figure 4E and F show different collections of gene sets. Figure 4E shows MSigDB and upregulated pathways. Figure 4F shows KEGG and downregulated pathways. The authors could show both the upregulated and downregulated pathways for a single collection of gene sets for consistency.

- In Figure 4E, “Uv response down” appears to the top upregulated gene set but it is not discussed by the authors. What is the interpretation of this result?

- In Figure 4F, Cell Cycle appears to be downregulated, however the authors showed that LINC00473 promotes cell proliferation. How do the authors reconcile this contradictory result?

- In Figures 4E and 4F, it appears that the color of the bars and the height the bars in the bar plot are both using the same information. For GSEA results it is typical to use the Normalized Enrichment Score (NES) or Enrichment Score (ES) in some fashion because p-value statistics are heavily dependent on gene set size.

- Figure 4I and 4J: instead of assessing the significance of gene set overlaps and showing Venn diagrams as in Figure 4I, it may be more useful to create gene sets and perform GSEA. I would recommend creating gene sets of different sizes from the RNA-Seq analysis of FLC vs NML (top 25, 50, 100, 200, 500 genes) and then run GSEA using the ranked list of LeGO-473ox relative to control genes as an input. An overlap of 188 genes might have some statistical significance, but 188 out of 1403 or 1667 genes suggests that there are many other differentially expressed genes that cannot be attributed to LINC00473 biology.

- In Figure 4J, the top downregulated gene set if TNFa-NFKB signaliing, but this is not discussed. What is the interpretation of this finding?

Reviewer #2: Fibrolamellar carcinoma is a rare cancer with limited options for patients due to late diagnosis and limited success on therapeutic strategies targeting the common DP-fusion established to be a driver of FLC. Therefore looking for downstream targets that could be developed is a very worthwhile effort. The establishment of LINC00473 as among the most highly upregulated genes in FLC (with limited number of samples albeit) and the subsequent work to carefully compare and dissect the expression LINC00473 from its host protein coding gene PDE10A is an important contribution. The demonstration in PDX models that patient-derived models of FLC, notably patient-derived xenograft (PDX) tissue as well as an FLC cell line established from this PDX, and found that DP fusion and LINC00473are concordantly elevated and siRNA studies to establish that Silencing of the DP fusion leads to LINC00473 suppression and siRNAs to LINC00473 impacts apoptosis, glycolysis and mitochondrial activity provides evidence for developing therapeutic strategies to impact the metabolic pathways driving FLC. In summary this is a well conducted study and more importantly provides evidence for role for LncRNAs as potential targets for cancer treatment.

Reviewer #3: The manuscript “DNAJB1-PRKACA fusion protein-regulated LINC00473 promotes tumor growth and alters mitochondrial fitness in fibrolamellar carcinoma” by Ma et al. utilizes functional genomics and in vivo models to characterize the role of the lncRNA LINC00473 in fibrolamellar carcinoma.

1. It does not appear from Figure 1A that LINC00473 is the most differentially expressed gene in the bulk RNA-sequencing analyses. Is there some interest in these other genes and what they might do?

2. What does RQV stand for in Figure 2, 3? It does not seem to be listed in the figure legend or paper.

3. Minor comment that the font on many of the figures is very difficult to read because it is quite small.

4. “These observations provide strong evidence that LINC00473 expression is driven by the DP fusion in FLC.” I think the data from Figure 2 really just demonstrates that LINC00473 expression in FLC requires DP fusion. This statement would be true if showing that the DP fusion specifically leads to LINC00473 expression. Demonstrating overexpression of the DP fusion in wild-type liver cells, then asking whether there is increased LINC00473 expression is a critical sufficiency experiment. The current experiment in the manuscript expresses LINC00473 in FLC cells that already have expression of LINC00473 – it is unclear to me what extra levels of LINC00473 does to the cells.

5. The proliferation experiments are purely based on cell counts – would be helpful to see a measure of viability using ATP-based readout assays such as Cell-Titer-Glo (CTG) for these measurements or using doubling time as a proxy rather than total cell number. Is this the viable cell count or just the total cell number?

6. In Figure 4A, it would be preferable to visualize the data using a volcano plot with log(fold change) and then replacing the mean normalized counts with significance (q-value/FDR). The mean normalized counts are generally not useful for these comparisons. It would helpful, moreover, to have the RNA-seq done in triplicate.

7. 8 days is quite short for a PDX experiment – do the authors have additional data showing that this is an ongoing sustained increase in tumor volume?

8. Do the authors have tumor tissue from these experiments that was processed for RNA-sequencing? This would give a better sense in an in vivo model what LINC00473 transcriptionally activates.

9. It would help to have a supplemental table of the most significant differentially expressed genes for RNA-seq studies.

Overall, I think the functional approaches utilized are robust and well controlled. While CRISPR-Cas9 approaches would be preferable, the siRNA approaches are reasonable. A major issue with the manuscript is that the figures are very difficult to read at 100% and for nearly all figures, I had to zoom in to 200-300% to read the figure legends. It would help the reader significantly for these legends to be more clear and refer specifically to the experiment (for example, if referring to qPCR data on DP fusion, to specifically show that in the figure). Otherwise, most of my concerns are around experimental details as detailed above.

**Have all data underlying the figures and results presented in the manuscript been provided?**

Reviewer #1: Yes

Reviewer #2: Yes

Reviewer #3: Yes

PLOS authors have the option to publish the peer review history of their article (what does this mean?). If published, this will include your full peer review and any attached files.

Reviewer #1: No

Reviewer #2: **Yes: **Preethi Gunaratne

Reviewer #3: No

---

## [Decision Letter · Decision Letter 1]

5 Feb 2024

Dear Dr Sethupathy,

Thank you very much for submitting your Research Article entitled 'DNAJB1-PRKACA fusion protein-regulated LINC00473 promotes tumor growth and alters mitochondrial fitness in fibrolamellar carcinoma' to PLOS Genetics.

Your revised manuscript has been evaluated by the three reviewers. All reviewers note significant improvement in your manuscript. There are several minor points of revision that are needed at this time. These points can be addressed through revising your manuscript text to discuss the research findings in the way suggested by the reviewers. Please adjust the language around GSEA/pathway analyses as suggested and clarify how the overexpression of LINC00473 compares to the overexpression of other genes in this disease. We will be happy to consider your revised manuscript. Please submit a point-by-point response to the reviewers' comments.  Assuming your responses are sufficient, we anticipate moving forward with your paper.

Yours sincerely,

John Prensner

Guest Editor

PLOS Genetics

Gregory Barsh

Editor-in-Chief

PLOS Genetics

Reviewer's Responses to Questions

**Comments to the Authors:**

Reviewer #1: It was my pleasure to review the revised manuscript entitled “DNAJB1-PRKACA fusion protein-regulated LINC00473 promotes tumor growth and alters mitochondrial fitness in fibrolamellar carcinoma” by Ma and colleagues. The authors took extraordinary measures to address reviewer comments, bolstering the use of the initial RNA-Seq cohort for discovery purposes and exploring possible hypothesis. All this reviewer’s original comments were addressed in the revised manuscript.

The revised manuscript includes numerous additional analyses, including gene set/pathway analyses that utilized enrichR and GSEA. I noted that in the rebuttal document and in the revised text the authors interpreted ranked lists of GSEA results.

I would urge the authors to exercise caution when interpreting GSEA results with p-values or adjusted p-values that are not significant. If statistically significance is borderline, one could increase the number of GSEA permutations to obtain a higher precision significant estimate. Otherwise, I would interpret the results as not enriched. I would not attempt to draw meaningful conclusions these results, which could occur due to chance, and instead state that the results were not significant. This would leave room for future follow-up studies exploring the role of LINC00473 in FLC.

I appreciate the extraordinary effort by the authors to provide a polished, high quality manuscript for publication.

Reviewer #2: The finding that LINC00473 as critical downstream driver of cell growth and survival in FLC harboring DNAJB1-PRKACA fusions both in PDX models and cell lines is an important contribution to understanding the gene networks that drive FLC. The study comprises the largest number of FLC samples analyzed through RNA-seq dataset published to date.

The work done to establish this especially with the addition of CRISPR- Cas9 to endogenously generate the DNAJB1-PRKACA fusion which was shown to significantly upregulated LINC00473 in HEK293-DP cells HepG2-DP cells with low or undetectable levels of the LncRNA provides strong evidence that LINC00473 is a downstream target of the onco-fusion.

Actionable fusions carrying one or both genes that have an FDA approved drug have significantly advanced cancer treatment. However, toxicities associated with the impact of these drugs on normal tissues expressing the wild-type form of the actionable gene has remains a major challenge. In that context finding critical drivers of oncogenic phenotypes downstream of actionable fusions is a very important next step. Functional studies to explore the impact of knock-down and upregulation of LINC00473 provides evidence for developing therapeutic strategies to impact the metabolic pathways driving FLC. RNAi-based therapies are currently in clinical trials however will take longer to get approved. In that context the finding that LINC00473 is also downstream of the fusion oncoprotein CRTC1-MAML2 in mucoepidermoid carcinoma (MEC) is mentioned as an opportunity for the work presented to have impact broader impact beyond FLC which is a rare cancer. In that context the manuscript would have strengthened in bringing in downstream drivers of LINC00473 as new druggable targets for FLC.

It is mentioned in the discussion that MUC5B and MUC14 established to be tumor-specific mucins in MEC and biliary tract carcinoma, are among the most highly expressed genes in FLC from RNA-seq. siRNA studies targeting MUC5B and MUC14 is able to rescue to oncogenic phenotype of either, in FLC cell lines carrying a DNAJB1-PRKACA-knock down and/or LINC00473.

Reviewer #3: Major points

1. Appreciate the authors’ explanation but I agree with Reviewer 1’s comments that LINC00473 does not appear to be the major downstream target of the DNAJB1-PRKACA fusion. It appears that the RNA-seq data was just used as more supporting data for the initial hypothesis. It would help if this was clearly stated in the manuscript in the description of the RNA-seq results (it appears to be #124 most upregulated gene in the list of differential expression..)

2. For the new data on tumors from LINC00473 cell implantation – MSigDB Hallmark and KEGG pathways are often somewhat generic. Was there any examination of the C3 regulatory gene sets for example that specifically look at transcription gene sets and miRNA gene sets. There are known miRNAs that LINC00473 regulates, and it would be a reasonable validation to see if those are reflected in the models generated.

3. I find it very odd that in the spreadsheet with the most significant differentially expressed genes for tumors that LINC00473 does not appear at all. That is somewhat concerning for these models.

I appreciate the authors’ responses to my prior comments. My continued concern is regarding the RNA-sequencing analysis, and the new dataset with tumor data is concerning given the lack of LINC00473 differential expression.

**Have all data underlying the figures and results presented in the manuscript been provided?**

Reviewer #1: Yes

Reviewer #2: Yes

Reviewer #3: Yes

PLOS authors have the option to publish the peer review history of their article (what does this mean?). If published, this will include your full peer review and any attached files.

Reviewer #1: No

Reviewer #2: **Yes: **Preethi Gunaratne

Reviewer #3: No

---

## [Editor Report · Decision Letter 2]

8 Mar 2024

Dear Dr Sethupathy,

We are pleased to inform you that your manuscript entitled "DNAJB1-PRKACA fusion protein-regulated LINC00473 promotes tumor growth and alters mitochondrial fitness in fibrolamellar carcinoma" has been editorially accepted for publication in PLOS Genetics. Congratulations!

Yours sincerely,

John Prensner

Guest Editor

PLOS Genetics

Gregory Barsh

Editor-in-Chief

PLOS Genetics

Comments from the reviewers (if applicable):

**Data Deposition**

http://datadryad.org/submit?journalID=pgenetics&manu=PGENETICS-D-23-01036R2

**Press Queries**

---

## [Editor Report · Acceptance letter]

18 Mar 2024

PGENETICS-D-23-01036R2 

DNAJB1-PRKACA fusion protein-regulated LINC00473 promotes tumor growth and alters mitochondrial fitness in fibrolamellar carcinoma 

Dear Dr Sethupathy, 

We are pleased to inform you that your manuscript entitled "DNAJB1-PRKACA fusion protein-regulated LINC00473 promotes tumor growth and alters mitochondrial fitness in fibrolamellar carcinoma" has been formally accepted for publication in PLOS Genetics! Your manuscript is now with our production department and you will be notified of the publication date in due course.

With kind regards,

Lilla Horvath

PLOS Genetics

On behalf of:
